# Mechanical control of neural plate folding by apical domain alteration

Miho Matsuda [1], Jan Rozman [2], Sassan Ostvar [3], Karen E. Kasza [3] & Sergei Y. Sokol [1]

Vertebrate neural tube closure is associated with complex changes in cell shape and behavior, however, the relative contribution of these processes to tissue folding is not well understood. At the onset of *Xenopus* neural tube folding, we observed alternation of apically constricted and apically expanded cells. This apical domain heterogeneity was accompanied by biased cell orientation along the anteroposterior axis, especially at neural plate hinges, and required planar cell polarity signaling. Vertex models suggested that dispersed isotropically constricting cells can cause the elongation of adjacent cells. Consistently, in ectoderm, cell-autonomous apical constriction was accompanied by neighbor expansion. Thus, a subset of isotropically constricting cells may initiate neural plate bending, whereas a 'tug-of-war' contest between the force-generating and responding cells reduces its shrinking along the body axis. This mechanism is an alternative to anisotropic shrinking of cell junctions that are perpendicular to the body axis. We propose that apical domain changes reflect planar polarity-dependent mechanical forces operating during neural folding.

Vertebrate neural tube closure (NTC) is a complex process that involves coordinated behaviors of cells that are controlled by hundreds of genes[1,2]. Neurulation starts with the specification of neuroectoderm and the formation of the neural plate. The neural plate bends at the medial hinge point to bring about the neural groove and triggers the elevation of the neural folds. The rising neural folds then bend at the paired dorsolateral hinge points and appose each other. Lastly, the folds fuse at the dorsal midline and lose contact with the surface non-neural ectoderm to complete NTC. Failure of NTC is a common birth defect and many responsible genes have been identified, including core components of planar cell polarity (PCP) signaling[2,3]. Despite extensive research efforts and long-standing debates in the field[3–6], elucidation of cellular and molecular mechanisms underlying NTC has been challenging.

Collective cell movements during neurulation are instructed by changes in cell shape and relative positions. Multiple cell behaviors, such as neighbor exchanges, apical constriction, and basolateral

protrusive activity, have been proposed as the cellular basis for NTC[7–12]. Whereas early studies suggested that apical constriction is a primary mechanism of tissue folding[3,13,14], other reports advocated the importance of neighbor cell exchanges for neural tube elongation[8,15]. To explain why the neural plate does not shorten in response to isotropic contractions of the apical surface[4], cell junctions were proposed to shrink anisotropically in response to PCP signaling[10]. Alternatively, the elongated shape of the neural tube may be instructed by tissue geometry, location of the constricting cell population, or affected by the underlying notochord. Due to the divergence of neurulation mechanisms in different species and the complexity of cell shape changes in distinct areas of the neural plate, the relative contribution of various cell behaviors to tissue folding remains poorly understood.

In this study, we evaluated changes in the apical domain of superficial cells in the *Xenopus* neural plate. The neural plate starts bending along specific lines of apically constricted cells at hinge points that are clearly visible in chick embryos[1,2] but less pronounced in

[1]Department of Cell, Developmental and Regenerative Biology, Icahn School of Medicine at Mount Sinai, New York, NY, USA. [2]Rudolf Peierls Centre for Theoretical Physics, University of Oxford, Oxford, UK. [3]Department of Mechanical Engineering, Columbia University, New York, NY, USA. ✉e-mail: sergei.sokol@mssm.edu

*Xenopus*[16]. At early neurula stages, we observed apically constricted cells interspersed with visibly expanded cells that were aligned with the AP axis, especially at the hinge areas. This apical domain heterogeneity required PCP signaling. A mechanical model simulation of the epithelial dynamics showed that the oriented cell elongation of the non-constricting cells can arise as a passive consequence of the presence of constricting cells. In vivo analysis of apical domain size in the ectoderm, in which actomyosin contractility has been activated, confirmed that apical constriction and expansion are mechanically coupled. Our study suggests that apical domain changes during neural plate folding involve PCP-dependent mechanical interactions between neuroepithelial cells.

## Results

### Heterogeneity of apical domains is a hallmark of neural plate folding

Vertebrate neurulation remains to be better understood at the cellular and molecular levels. Recent time-lapse imaging studies reported distinct frequencies of cell intercalations and apical constriction behaviors in the posterior and the anterior regions of *Xenopus* neural plate[7,8,17]. Approximately one third of superficial layer cells undergo neighbor exchanges leading to mediolateral narrowing and anteroposterial elongation of the posterior NP[8]. Our analysis documented a similar frequency of cell intercalations in the posterior NT (25.4%, $n = 284$, between stages 13 and 15). Such neighbor exchanges were less frequent in the anterior neural plate, in which apical constriction was reported to be the predominant mode of cell shape change in neuroepithelial cells[7,17]. Moreover, apically constricted cells are abundant at the dorsolateral and medial hinges of the folding posterior neural tube[1,3]. These observations suggest that apical domain changes contribute to neural plate bending more than previously appreciated.

En face view of phalloidin-stained neurulae showed that the whole neural plate is enriched with F-actin (Fig. 1a), possibly reflecting elevated contractility as compared to the non-neural ectoderm. Binary segmentation revealed variable morphology of apical domains in the cells throughout the neural plate (Fig. 1a, a'). The most pronounced variability was observed in the hinge areas of stage 15 neurulae (Fig. 1b, d). Of note, the majority of cells in the neural plate were oriented along the anteroposterior (AP) body axis (Fig. 1c). We found that both medial and dorsolateral hinges contained clearly constricted cells that were alternating with the cells elongated along the AP axis (Fig. 1b, d). Quantitative analyses showed increased variability of both apical domain size and cell aspect ratio in the hinge area of the neural plate, compared to those in the non-hinge area and the stage 11 embryonic ectoderm (Fig. 1e, f, Supplementary Table 1). We conclude that the neural plate acquires considerable cell heterogeneity by the onset of folding, consistent with a previous study[7]. Together, these findings suggest that the observed apical domain heterogeneity plays a role in neural plate folding.

### PCP signaling is required for apical domain heterogeneity in the neural plate

Coordinated cell polarization in the epithelial plane, known as planar cell polarity (PCP), commonly underlies cell shape changes during morphogenesis, including vertebrate NTC[18–21]. PCP in the neural plate can be visualized by nonuniform distribution of core PCP protein complexes and requires the function of the conserved core PCP protein Vangl2[22,23]. Our previous studies suggested that PCP signaling is required for apical constriction in the blastopore during *Xenopus* gastrulation and during NTC[24,25]. Therefore, we asked whether PCP signaling regulates the apical domain heterogeneity in the neural plate. Unilateral knockdown of *vangl2* using previously characterized *vangl2* morpholino oligonucleotide (MO)[24,26] with GFP as a lineage tracer reduced apical domain heterogeneity in the MO-injected half of the neural plate (Fig. 2a, b', Supplementary Fig. 1).

Since F-actin is likely to play a role in NP folding, we quantified phalloidin staining in wild-type and *vangl2* morphant neural plates. In wild-type neural plates, mediolaterally oriented junctions that are perpendicular to the AP axis were shorter and had stronger phalloidin staining in the posterior neural plate (Fig. 2, Supplementary Fig. 2), consistent with previously described enrichment of F-actin at shorter junctions[7,8]. In *vangl2*-depleted cells, the overall intensity of junctional F-actin was lower than in the uninjected cells (Fig. 2b, b', and Supplementary Figs. 1c, c', 2). The observed reversed correlation between the junction length and F-actin intensity and the difference in overall F-actin intensity between wild-type and *vangl2* morphant cells may be important for NP bending.

Of note, the effect of *vangl2* knockdown was visible in both posterior (Fig. 2b–d) and anterior neural plate (Supplementary Fig. 1c-e), but absent in the non-neural ectoderm (epidermis)(Supplementary Fig. 1f, g). Our findings indicate that the cell heterogeneity develops in the neural plate under the control of PCP signaling. Importantly, mosaic depletion of Vangl2 in the neuroectoderm increased apical domain size (Supplementary Fig. 3). These effects were cell-autonomous, i. e. observed in the cells depleted of Vangl2. We confirmed that the GFP lineage tracer was properly targeted to the ectoderm and largely absent from the mesoderm in these experiments (Supplementary Fig. 4a, b") and the knockdown minimally affected the length of the injected embryos (Supplementary Fig. 1a, b', c). These observations suggest that Vangl2 regulates apical domain heterogeneity, possibly by influencing local rather than global mechanical forces.

### Cell elongation can arise passively from the presence of interspersed contractile cells

The heterogeneity of apical domains in the neural plate indicates that only a subset of the cells undergo apical constriction. We asked whether the observed cell behavior in the tissue can be explained as a mechanical consequence of apical contractility. We therefore analyzed a 2D vertex model[27–29] of the neural plate epithelium. In a 2D vertex model, each cell is represented as a polygon and defined by the position of its vertices. An energy is chosen for the model tissue, and the vertices are then assumed to follow overdamped dynamics towards an energy minimum. In this case, we use the usual area-and-perimeter elasticity energy, which sets a target area and perimeter value for each cell[27]. The energy, therefore, reads

$$e = \sum_{c}^{N} \left[ \left( a^{(c)} - 1 \right)^2 + k_P \left( p^{(c)} - p_0^{(c)} \right)^2 \right], \qquad (1)$$

where the sum runs over all $N$ cells, $a^{(c)}$ and $p^{(c)}$ are the area and the perimeter, respectively, of the $c$-th cell, whereas $p_0^{(c)}$ is its target perimeter; $k_p$ is the perimeter elasticity modulus (this is the dimensionless form of the energy; see Methods). We set $k_p = 1$ and $p_0^{(c)} = 3.7$, just below the perimeter of a regular hexagon of area 1, and started the simulation with a regular hexagonal lattice. A $N_x$ by $N_y$ rectangular region (blue border in Fig. 3a) represents the posterior region of the neural plate. Here, $N_x$ corresponds to the length along the AP axis, and $N_y$ the length perpendicular to the AP axis. The model posterior neural plate is embedded in a $N'_x$ by $N'_y$ simulated tissue. Unless stated otherwise, we set $N_x = 60$, $N_y = 20$, $N'_x = 100$ and $N'_y = 60$.

At the start of the simulation, each cell in the neural plate region has a probability $P_c$ of becoming apically constricting (Fig. 3a). We modeled constriction by setting the target perimeter of constricting cells to 10% of that of the non-constricting cells. Figure 3b shows the tissue shape after relaxation at time $t = 2000$, with red lines indicating the direction of elongation for the non-constricting cells, whereas Fig. 3c shows the distribution of angles between cell elongation and the AP axis. This simulation suggests that the anisometric shape of the constricting region is sufficient for the non-constricting cells to favor

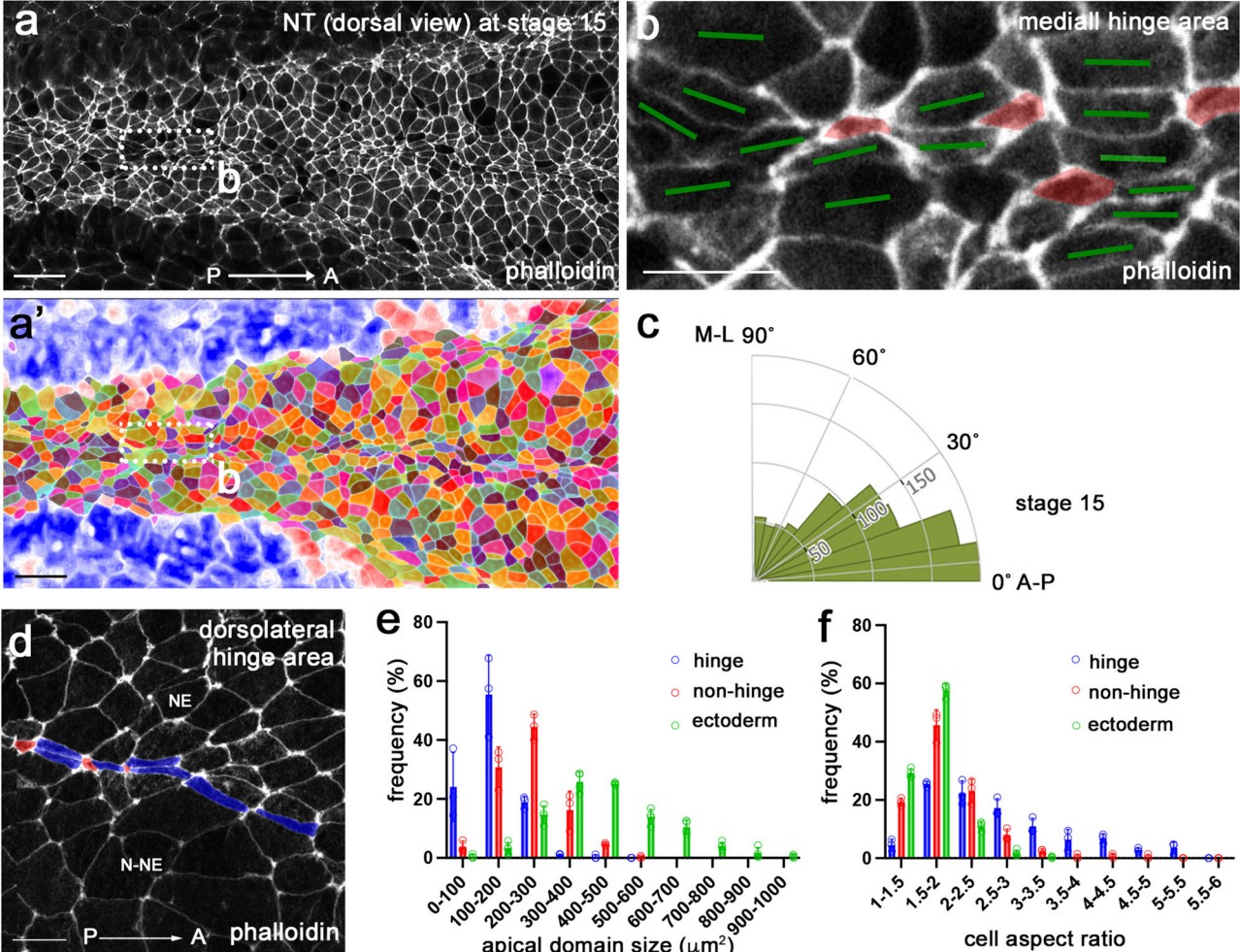

**Fig. 1 | Neural plate hinge cells contain heterogeneous apical domains. a, b'**
Dorsal view of the middle area of a control *Xenopus* neural plate stained with
phalloidin (**a**) and the segmented image (**a'**) at stage 15. The rectangular area in (**a**),
corresponding to the medial hinge, is enlarged in (**b**). **b** Cells with small apical
domain (red) are interspersed with large cells elongated along the AP axis (green
bars). **c** Rose plots show the orientation of the cells relative to the anteroposterior
(AP) axis. *n* = 929 cells. Data from three stage 15 embryos are combined. **d** In the
dorsolateral hinge region, cells with small apical domain (red) are adjacent to
elongated cells (blue). Neuroepithelium (NE) and non-neural ectoderm (N-NE) are
in the upper and lower parts of images, respectively. The histogram of apical
domain size (**e**) and cell aspect ratios (**f**) of cells in stage 11 embryonic ectoderm

(green), and cells in the medial or dorsolateral hinge (blue) and non-hinge (red)
areas of stage 15 neural plate. Four-to-five rows of cells at the dorsal midline were
considered the medial hinge area. The non-hinge areas exclude four-to-five rows of
cells from the medial and dorsolateral hinges. Data (means ± s.d.) are combined for
three stage 15 embryos and four 11 stage embryos, representative of three inde-
pendent experiments. Stage 15 hinge, *n* = 255 cells; stage 15 non-hinge, *n* = 426
cells; stage 11 ectoderm, *n* = 1218 cells. The experiments were repeated at least
three times. Coefficients of variation (CV) are shown in Supplementary Table 1.
One-way ANOVA Kruskal-Wallis test. One-sided. $p < 0.0000000001$ (**e**).
$p < 0.0000000001$ (**f**). Scale bars are 50 μm in (**a, a'**), 20 μm in (**b**) and (**d**).

elongation along the region's long axis. If only a fraction of cells in the
neural plate constrict, the tissue shrinks less than when all cells con-
strict (Fig. 3d). Note that in the model, the non-constricting cells
elongate but do not usually expand; their final average area is ~9%
lower than in the initial condition.

The underlying physical mechanism for cell elongation is there-
fore similar to the one proposed to explain why the mesodermal
domain during *Drosophila* ventral furrow formation contracts much
less along the AP axis than perpendicular to it[30]. A rectangular con-
tractile domain in a flat 2D elastic plate will contract more along the
short than the long axis of the domain[31,32]. A similar mechanism has
recently also been studied in 3D[33]. Importantly, this also implies that
the external tissue outside the neural plate counterbalancing the
deformation is an important factor in the observed cell elongation[31].
The difference in the present model when compared to the *Drosophila*
models above is that there are now two populations, with only one
generating the contractility. A related behavior has also been reported

and modeled during the slow phase of *Drosophila* ventral furrow
formation[34,35].

As the cell alignment with the AP axis depends on the interplay
between the constricting cells and the passive response of the external
tissue, the results are sensitive to the size and shape of the external
region outside the model neural plate. In the simulations discussed
above, we set $N'_x$ and $N'_y$ so that there is an equivalent number of cells
between the model neural plate and the boundary of the simulation
box along and perpendicular to the AP axis. Changing $N'_y$ to 34 while
keeping $N'_x = 100$ so that the aspect ratio of the simulated tissue and
the model neural plate are similar removes the alignment effect
(Supplementary Fig. 5a, b, g). Inversely, using a square $N'_x = N'_y = 100$
simulated tissue leads to more pronounced alignment of non-
constricting cells with the AP axis (Supplementary Fig. 5c, d, g).
Moreover, further increasing the total number of cells (e.g., to
$N'_x = N'_y = 180$) then reduces the extent of alignment compared to the
initial case (Supplementary Fig. 5e, f).

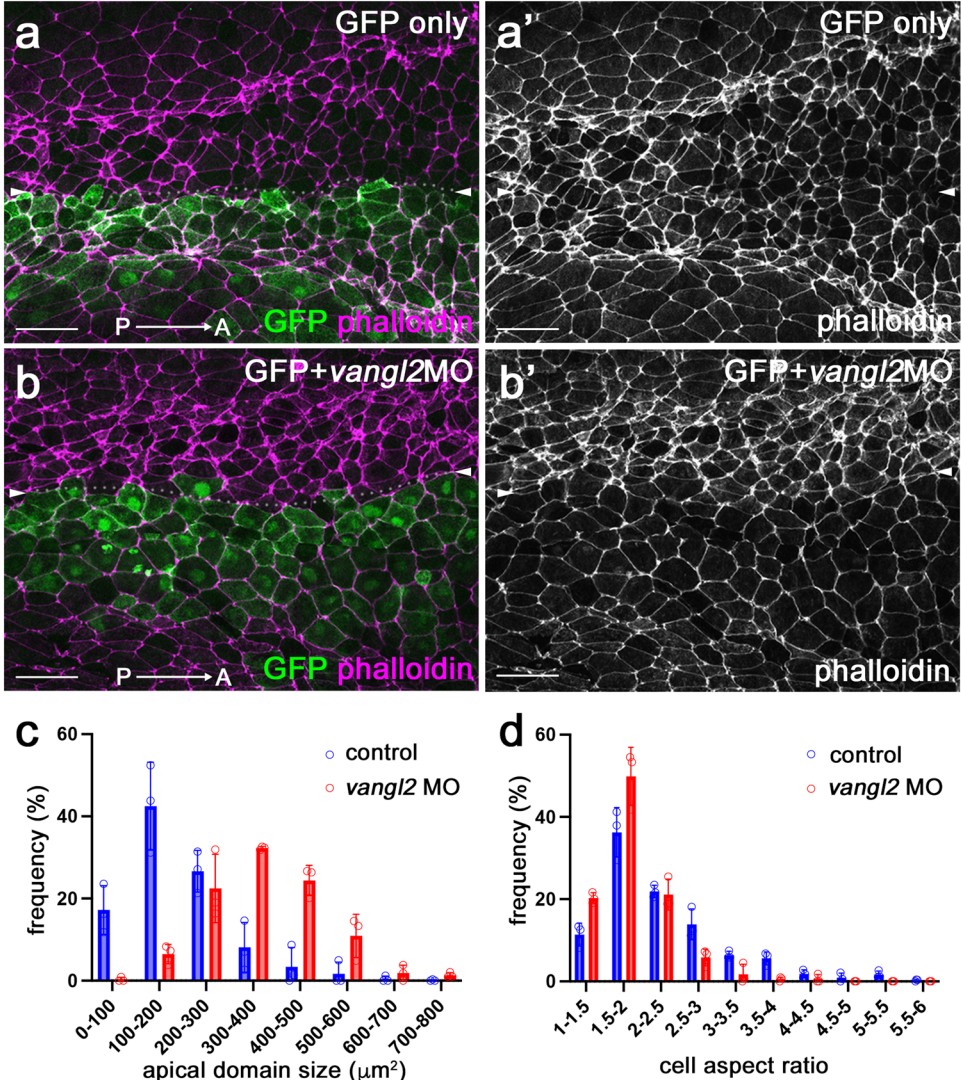

**Fig. 2 | PCP signaling is required for apical domain heterogeneity in the neural plate. a, b'** Representative images of control GFP-injected embryo and *vangl2* MO-injected embryo (the injected side marked by GFP). Dorsal view. Control GFP RNA (**a, a'**), or GFP RNA and 10 ng *vangl2* MO (**b, b'**) were injected into four-cell embryos targeting the presumptive neural plate. Control, *n* = 6 embryos; *vangl2* MO, *n* = 9 embryos. (**c, d**) The histogram of apical domain size (**c**) and cell

aspect ratio (**d**) of uninjected and *vangl2* MO-injected cells in (**b, b'**). The hinge and non-hinge regions were quantified together. Data (means ± s.d.) was quantified with *n* = 853 cells from three control embryos and *n* = 359 cells from three *vangl2* MO-injected embryos. This represents three independent experiments. The Kolmogorov-Smirnov test. Two-sided. *p* < 0.00000000001 (**c**). *p* = 0.000000004 (**d**). CVs are shown in Supplementary Table 1. Scale bars are 50 µm in (**a, b'**).

We also ran simulations with different $N_x$ and $N_y$, changing the aspect ratio of the constricting region $\alpha = N_y/N_x$, but keeping the product $N_x N_y$ similar to the above case. The size of the total simulated tissue was set to $N'_x = N_x + 40$ and $N'_y = N_y + 40$. We found that cells are on average aligned closer to the *x* axis for $\alpha < 1$, whereas they align closer to the *y* axis for $\alpha > 1$ (Fig. 3e). As the anisometry of the region (i.e., aspect ratio either increasing or falling away from 1) increased, cells also became more elongated (Fig. 3f).

We next considered a more detailed model that includes a higher fraction of constricting cells in the two hinges. We separated the neural plate into two 3-cell-wide regions, representing the hinges, which are separated by ~14 cells. Cells in the hinges have a probability $P_h = 0.5$ of constricting, whereas those in the region between hinges only constrict with a probability $P_c = 0.2$ (Supplementary Fig. 6a). Interestingly, we found that in this case only cells in the hinges elongate along the AP axis, whereas the remaining cells in the neural plate do not have a notable preferred direction of elongation (Supplementary Fig. 6b–d). Thus, the geometry of the shrinking tissue depends on the frequency and distribution of apically constricting cells.

Lastly, we used a 3D vertex model[36–38] (Fig. 3g–j, see Methods) to test whether the constriction of only a fraction of cells is still sufficient for the formation of the furrow. Here, we are only interested in the hinge, so we limited the region in which cells have a probability of constricting to a ~3-cell-wide patch (blue border in Fig. 3g), with periodic boundary conditions along the AP axis. As in the 2D model, constriction was modeled by reducing the target perimeter of the apical cell side to 10% of its initial value. The resulting tissue morphology after relaxation at $P_c = 0.5$ is shown in Fig. 3h, i: the constriction of only ~50% of cells is indeed sufficient for the formation of at least a shallow furrow. However, the furrow is deeper in a model tissue in which all hinge cells constrict (Fig. 3j). Altogether, our simulations allow us to propose that cell elongation in the neural plate is a passive mechanical consequence of cell constriction. Lastly, we chose to focus on minimal models of the interplay between cell constriction and the consequent elongation. Therefore, additional mechanisms included in other vertex models of neural tube formation[39,40] are not considered here.

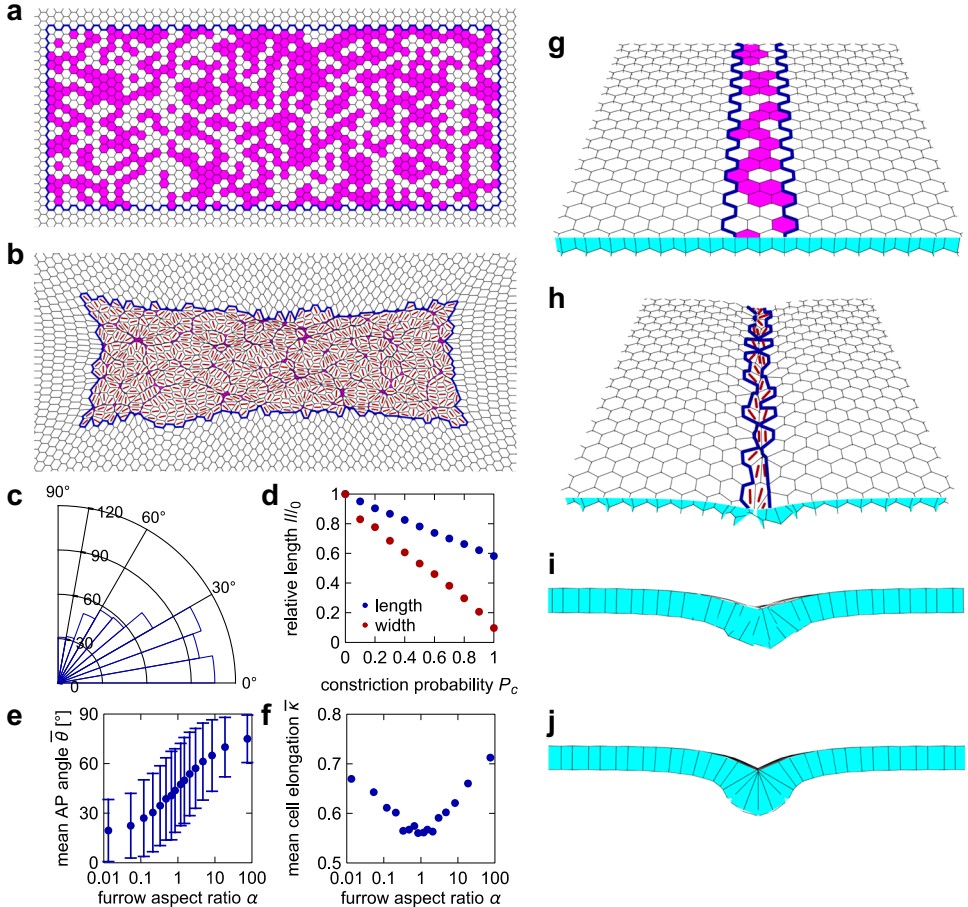

**Fig. 3 | Mechanical model shows that contractile cell subpopulation leads to cell elongation and alignment. a–f** 2D vertex model of the neural plate. **a** Zoom on the neural plate region of the vertex model initial condition. Blue border outlines the posterior neural plate region of the model tissue. Apically constricting cells are in magenta (tissue shown for $P_c = 0.5$). **b** Model tissue from (**a**) after relaxation at $t = 2000$. Red lines show the direction of cell elongation. **c** Distribution of angles between the AP axis (horizontal line) and cell elongation direction of non-constricting neural plate cells for the model posterior neural plate in (**b**); 0° corresponds to perfect alignment. **d** Central length (along AP) and width (perpendicular to AP) of the model tissue after relaxation as a function of the probability of cell constriction $P_c$; values are normalized by length and width if no cells constrict. **e** Mean angle with the AP axis as a function of the constricting

region aspect ratio $\alpha$. Error bars indicate the standard deviation between all non-constricting cells in one instance of the model tissue for each $\alpha$ (number of cells $n = 663, 626, 614, 616, 608, 605, 590, 610, 609, 587, 600, 599, 604, 595, 594, 592$ in order of increasing $\alpha$). **f** Mean cell elongation as a function of the furrow aspect ratio. See Methods for details. **e** and **f** are for $P_c = 0.5$. **g–j** Furrow formation in a 3D vertex model simulation of the hinge area. **g** Initial condition for the 3D vertex model simulation. Hinge region is outlined by a blue border and constricting cells are shown in magenta (tissue shown for $P_c = 0.5$). **h** Model tissue from (**g**) after relaxation at time $t = 5000$. Red lines show the direction of elongation for the apical side of the cells. **i** Cross-section view of the model tissue in (**h**), showing the formation of a shallow furrow. **j** Cross-section view of a model tissue with $P_c = 1$.

## Lmo7-expressing ectoderm as a model of apical domain heterogeneity

Our in silico model predicts that mechanical forces are sufficient to instruct cell elongation and orientation in a tissue with a subset of dispersed contractile cells. Testing this hypothesis in the neural plate is challenging due to its complex tissue dynamics and patterning events. We therefore examined apical domain changes in constricting gastrula ectoderm, a tissue from which the neural plate originates.

We induced apical constriction in *Xenopus* ectoderm using Lmo7, a myosin II-interacting protein[41]. The injection of *gfp-lmo7* RNA in two opposing blastomeres of 4–8-cell embryos resulted in ectoderm hyperpigmentation (Fig. 4a, b) that commonly accompanies apical constriction in *Xenopus* ectoderm[41]. Notably, the majority (80–90%) of injected embryos showed coordinated cell movements (Movie 1) and developed a pigmented furrow between the two injection sites (Fig. 4b, c). Importantly, GFP-Lmo7 cells in the furrow were aligned (Fig. 4d, e). Similar hyperpigmentation and furrow formation have been observed in the ectoderm cells after the induction of apical constriction by the actin-binding protein Shroom3[42,43]. Since both

Lmo7 and Shroom3 induce apical constriction and participate in NTC, the cellular mechanisms underlying furrow formation in the apically constricting ectoderm may be akin to those operating in the neural plate. We therefore used apical constriction in ectoderm to study the processes underlying neural plate folding.

## Apical domain dynamics in Lmo7-expressing ectoderm

We monitored apical domain dynamics by live imaging of embryos microinjected into the four animal blastomeres with *gfp-lmo7* RNA (100–200 pg) (Fig. 5a). At this dose, GFP-Lmo7 was uniformly expressed in the ectoderm and the expressing cells had similar apical domain size at the beginning of imaging at stages 10.5–11 (Fig. 5b). By the end of imaging (3.5–4 h), however, some cells apically constricted (AC cells, red in Fig. 5b–d') whereas others exhibited continuous apical expansion (AE cells, blue in Fig. 5b–d')(Movie 2). We confirmed this observation by quantifying apical domain size dynamics of individual cells in several GFP-Lmo7 embryos (Fig. 5e, g, Supplementary Fig. 7a–c). Of note, AC cells were often located adjacent to AE cells (Fig. 5b, c). By contrast, apical domain size did not significantly change in control

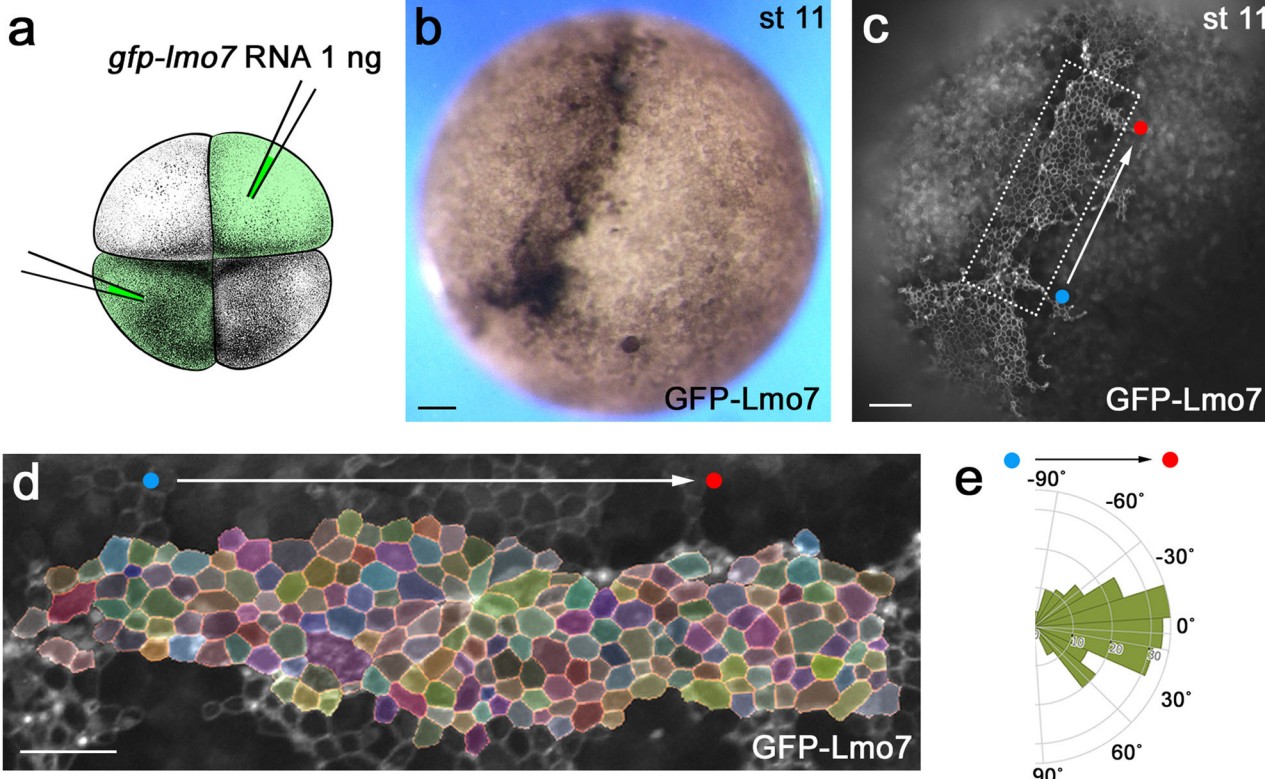

**Fig. 4 | Lmo7-expressing ectoderm as a model of apical domain heterogeneity.**
**a** Scheme of the experiment. GFP-Lmo7 RNA (1 ng) was injected in two oppositely localized animal blastomeres in four-cell stage embryos. Adapted from *Xenopus* illustrations © Natalya Zahn (2022), Xenbase (www.xenbase.org RRID:SCR_003280)[75]. **b** Representative GFP-Lmo7-expressing embryo at stage 11. Hyperpigmentation was observed in 90–95% embryos (*n* > 100). **c** Still image from Movie 1 shows GFP-Lmo7 fluorescence from the embryo in (**b**). **d** Segmented image of the rectangular area in (**c**). **e** Representative rose plot from *n* = 792 cells from one embryo depicts cell orientation with respect to the injection axis (white arrow connecting red and blue dots in **c**–**e**). The data represent at least three experiments. Scale bars are 100 μm in (**b**, **c**) and 50 μm in (**d**).

cells that expressed GFP-ZO1 (Fig. 5f, g, Supplementary Fig. 8) (Movie 3) or membrane-tethered GFP-CAAX (Supplementary Fig. 7a, c).

We describe the emergence of the striking heterogeneity of apical domains in Lmo7-expressing epithelium that is similar to the one visible in the neural plate. Although the origin of this heterogeneity in the neuroectoderm is unknown, our data suggest that it reflects the mechanical interactions between neighboring cells. To understand whether the selection of the contricting and nonconstricting cell populations is stochastic or involves some prepattern, we carried out an in situ hybridization for *lmo7*. Whereas *lmo7* transcripts were largely missing in ventral ectoderm, we observed punctate staining in the dorsal ectoderm (corresponding to neuroectoderm) at the end of gastrulation (Supplementary Fig. 9). This result is consistent with a prepatterning process that selects the constricting cell population in the neural plate. Whereas our observations are similar to previous work suggesting that local strains are sufficient to initiate NP folding in neural explants[44], our experiments do not exclude a role of non-neural tissue-derived forces in this process.

### Cells mosaically-expressing Lmo7 are predominantly constricting

Uniform Lmo7 expression in *Xenopus* ectoderm triggered the appearance of both AC and AE cells. This observation may be explained by direct effects of Lmo7 on apical constriction and apical expansion. Alternatively, AE cells may form as a passive response to the AC cells as predicted by our vertex model. To test these possibilities, we compared apical domain dynamics in embryos expressing Lmo7 uniformly ("sheet" in Fig. 5a) or mosaically ("isolated", Fig. 6a

and Supplementary Fig. 10a, b). We confirmed that the uniform expression of Lmo7 triggered both apical constriction and expansion (Fig. 6b, c', h, j) (Movie 4), whereas the majority of "mosaic" Lmo7 cells were constricting (Fig. 6d–g', i, j, Supplementary Fig. 10c-c", f, g) (Movie 5). Note that due to a shorter time-lapse imaging period, the frequencies of AC and AE cells in a "sheet" are smaller as compared with 4–5 h imaging (compare Figs. 5g, 6j, and Supplementary Fig. 7c). By contrast, control cells expressing myr-BFP did not show a significant change in the apical domain size (Supplementary Fig. 10d, d", e, g). These results suggest that the primary function of Lmo7 is to induce apical constriction. Since the AE cells were often found adjacent to the AC cells (Figs. 5b–6d', b, c'), the increase of their apical surface was likely driven by the pulling force of neighboring AC cells.

### Apical domain heterogeneity in ectoderm cells expressing Shroom3

We next asked whether the observed modulation of apical domain size was specific to Lmo7 or shared by other apical constriction regulators. Shroom3 is a known to induce apical constriction and is essential for vertebrate NTC[42,43]. Similar to Lmo7 cells, the ectoderm cell population uniformly expressing Shroom3 had increased aspect ratio and the cells were oriented relative to the axis connecting two injection sites ("line of constriction", Supplementary Fig. 11a–e).

We also examined apical domain size in ectoderm cells mosaically-expressing Shroom3 (Fig. 7a). We observed that these cells had smaller apical domains (Fig. 7b–d) and a wider range of cell aspect ratio compared to neighboring wild-type cells (Fig. 7b, c', e). These wild-type cells increased their apical domain size in response to adjacent Shroom3 cells (Fig. 7c, c'). These findings support the view that

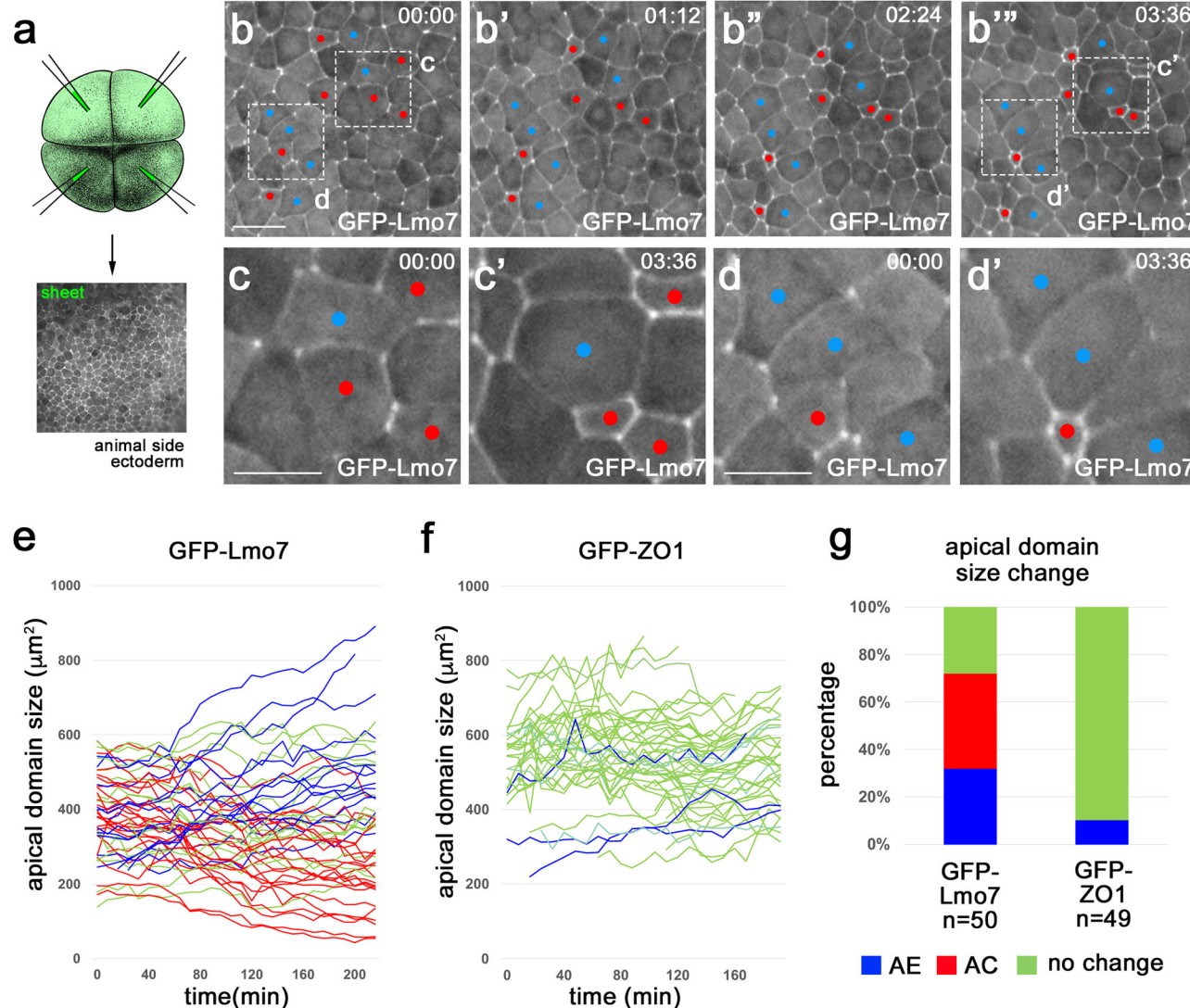

**Fig. 5 | Apical domain heterogeneity in ectoderm cells expressing Lmo7.**
**a** Scheme of the experiment. GFP-Lmo7 (100 pg) or GFP-ZO1 RNA (150 pg) was injected into four animal blastomeres of 4–8-cell embryos for live imaging at stage 11. Uniform fluorescence has been confirmed in stage 11 ectoderm. Adapted from *Xenopus* illustrations © Natalya Zahn (2022), Xenbase (www.xenbase.org RRID:SCR_003280)[75]. **b-d′** Time-lapse imaging of GFP-Lmo7 embryos for ~4 h. Apically constricting (AC) and expanding (AE) cells are marked by red and blue, respectively. Areas in (**b**) and (**b′′′**) are enlarged in (**c**, **d**) and (**c′**, **d′**), respectively. Quantification of apical domain dynamics in GFP-Lmo7 (**e**) and GFP-ZO1 (**f**) cells in one representative embryo. Each line represents apical domain size changes of individual cells for ~4 h. The cells were scored as AC (red) or AE (blue) if they had more than 20% decrease or increase in their apical domain size, respectively. The remaining cells were designated as 'no change' (green). $n = 50$ cells for GFP-Lmo7. $n = 49$ cells for GFP-ZO1. **g** Frequencies of cells with different changes in apical domain size are shown for embryos in (**e**) and (**f**). The Freeman-Halton extension of Fisher's exact test. Two-sided. $p = 2.383$ E-13. These experiments were repeated 3–5 times. Scale bars are 40 µm in (**b**) and 20 µm in (**c**, **d**).

the coordinate regulation of apical domain size is a general property of epithelial morphogenesis.

Based on our observations, we propose a model in which the dispersed population of isotropically constricting cells is sufficient to drive neural plate bending, whereas the coexistence of the shrinking and elongating cells at the hinges combined with mechanical constraints in the tissue leads to alignment of elongated cells and reduces tissue shrinking along the AP axis (Fig. 7f).

## Discussion

This study investigated the origin of apical domain heterogeneity of the superficial cell layer in the vertebrate neural plate and its contribution to NTC. Mediolateral and radial cell intercalations of the superficial and deep neural plate cells are known to play a role in the elongation and the final morphology of the NT[9,45,46]. Indeed, different papers including this work report 20–50% superficial cells undergoing

neighbor exchanges at stages 13–15[7,8,17]. Nevertheless, how NP elongation caused by superficial cell intercalations contributes to folding is currently unclear. Our work primarily focused on apical constriction that is manifested throughout the length of the neural plate, most notably at the medial and dorsolateral hinges. We observed that apically constricted cells were interspersed with cells that apically expanded along the AP neural axis, especially in hinge areas. Simulations of the tissue constriction using 2D and 3D vertex models fit well with our experimental observations. We propose that neural folding relies on intrinsic mechanical forces arising in the apically constricting subpopulation of neuroepithelial cells. Our hypothesis suggests that the neural plate is under anisotropic stress, with the higher tension along the anteroposterior body axis. Supporting this view, a cut placed orthogonally to the AP axis produced a wound that expanded its perimeter faster and healed slower than the wound that was parallel to the AP axis[47]. Importantly, laser ablations in the *Xenopus* neural plate

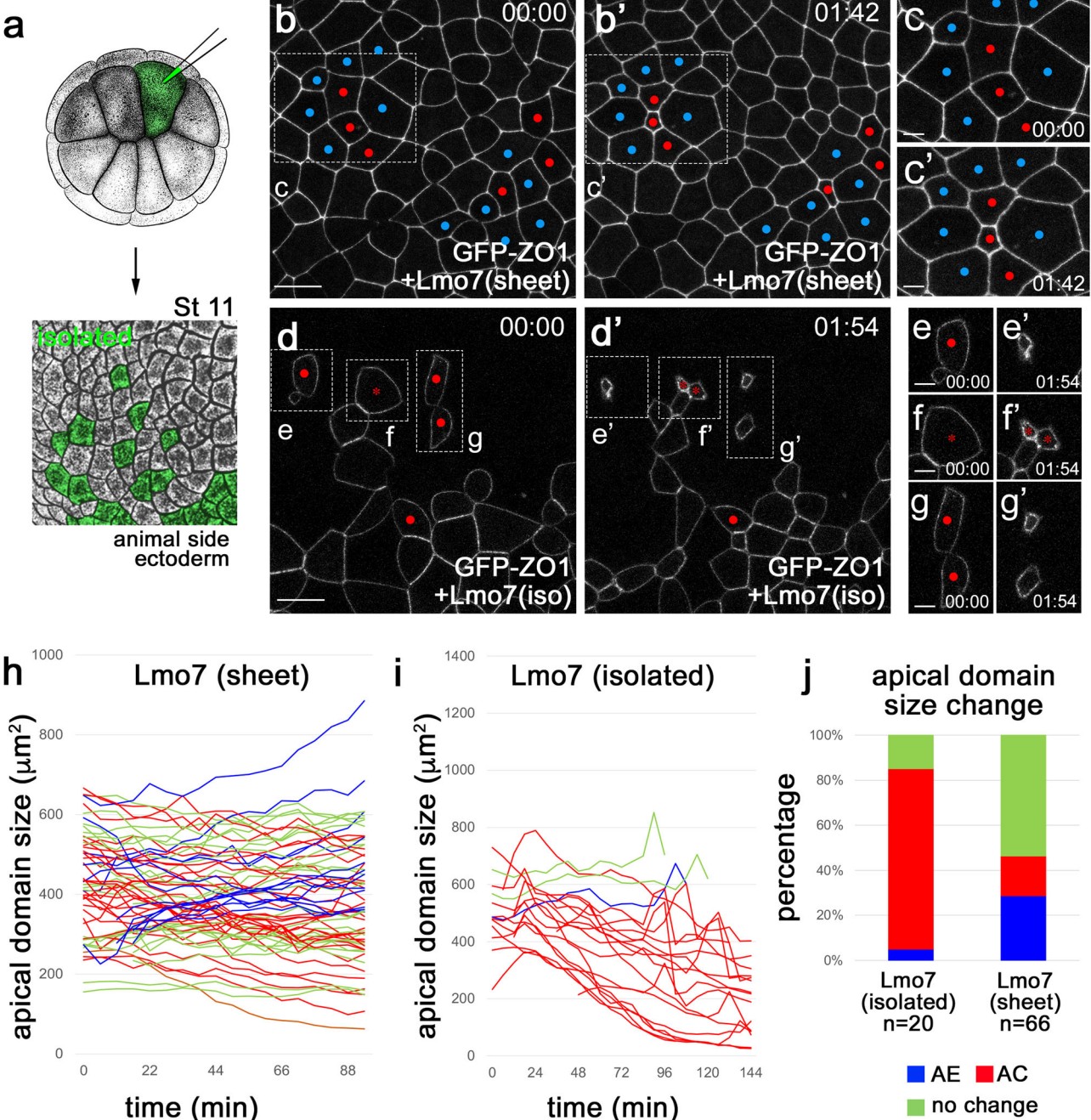

**Fig. 6 | Mosaic expression of Lmo7 causes apical constriction. a** Scheme of the experiment. GFP-ZO1 RNA (100 pg) and Flag-Lmo7 RNA (100 pg) were coinjected into one ventral blastomere of 16-cell stage embryos. Adapted from *Xenopus* illustrations © Natalya Zahn (2022), Xenbase (www.xenbase.org RRID:SCR_003280)[75]. **b–g′** Representative images of GFP-ZO1 cells co-expressing Flag-Lmo7 in "sheet" (**b, c**) or in "isolated" from Movies 4 and 5. Time-lapse imaging was initiated at stage 11. AC and AE cells are marked by red and blue, respectively. Asterisks show cells undergoing mitosis during the imaging. Rectangular areas in (**b, b′** and **d**), d′ are enlarged in (**c, c′**) and (**e–g′**), respectively. Scale bars are 30 μm in (**b, d**) and 10 μm in (**c, e–g**).

Experiments are repeated 3–5 times. Apical domain dynamics in the "sheet" (**h**) and "isolated" ectoderm (**i**). Each line represents apical domain size changes of an individual cell over 1.5–2 h. AC, AE, and "no change" cells (see Fig. 5 legend) are shown in red, blue, and green, respectively. Quantification was based on $n = 66$ cells from one Flag-Lmo7 (sheet) embryo and $n = 20$ cells from three Flag-Lmo7 (isolated) embryos. **j** Frequency (%) of AC, AE and "no change" cells in (**h** and **i**). These experiments were repeated 3–5 times. The Freeman–Halton extension of Fisher's exact test. Two-sided. $p = 0.00242$.

resulted in a similar conclusion[48]. Whereas our results are consistent with apical constriction being a driver of initial neural plate bending, we do not exclude significant contributions of the inner layer cells and forces coming from non-neural tissues to neural tube closure as proposed by other studies[3,9,39,49].

To further understand the mechanisms underlying cell shape changes, we monitored apical constriction induced in the superficial ectoderm by Lmo7 and Shroom3. Experimental induction of AC and AE cells in the ectoderm caused cell alignment relative to the apical constriction axis and triggered furrow formation, which suggests that apical domain heterogeneity contributes to epithelial folding. We find that a subpopulation of dispersed contractile cells is sufficient to trigger epithelial bending and that the apically expanded cells react to the pulling forces of their constricted neighbors. Thus, the elongated

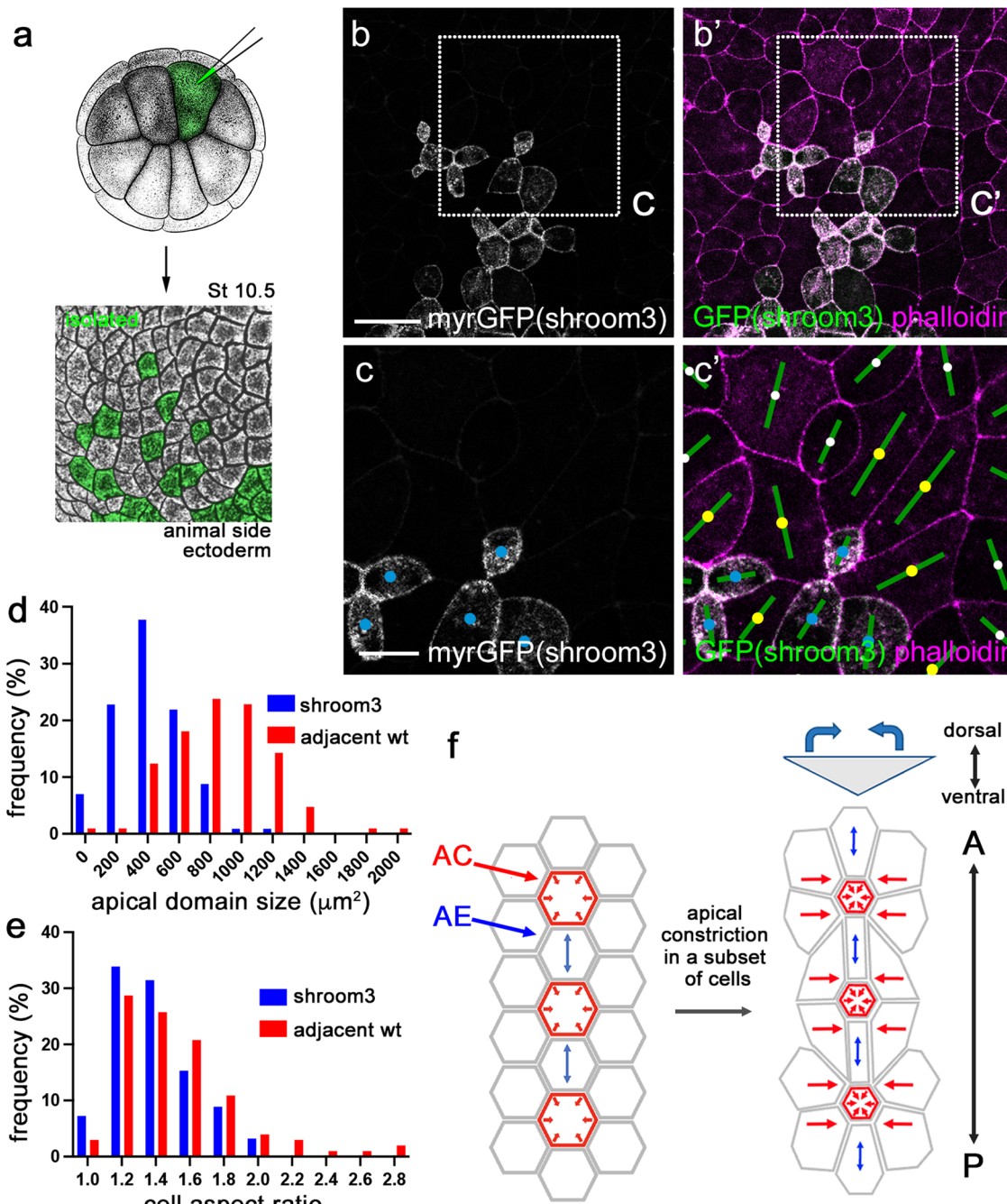

**Fig. 7 | Modulation of the apical domain by Shroom3. a** Scheme of the experiment for G–I. Shroom3 (80 pg) and myrGFP (50 pg) RNA were co-injected into one animal blastomere of 16-cell stage embryos. Stage 10.5 embryos were fixed, stained with phalloidin and the animal pole ectoderm was imaged. Adapted from *Xenopus* illustrations © Natalya Zahn (2022), Xenbase (www.xenbase.org RRID:SCR_003280)[75]. **b**, **c′** Shroom3 cells (yellow) with adjacent wild-type cells (blue) are shown. Rectangular areas in B and B′ are enlarged in (**c**) and (**c′**). Cell orientation is shown by green bars. **d** Apical domain size of Shroom3 cells and the adjacent wild-type cells. **e** Cell aspect ratios for Shroom3 cells and the adjacent wild-type cells. *n* = 114 (Shroom3 cells) and *n* = 105 (adjacent wild-type cells) from 5 embryos were analyzed for (**c**) and (**e**). The Kolmogorov-Smirnov test. Two-sided. *p* < 0.000000001 (**d**). *p* = 0.0266 (**e**). Experiments were repeated three times. **f** Model of neural plate bending. A subset of cells at neural plate hinge lines are selected to undergo PCP-dependent isotropic apical constriction (red arrows). Due to the geometry of the neural plate, the adjacent cells passively respond to the pulling forces exerted by their constricting neighbors by elongating along the anteroposterior axis (blue arrows). This cell alignment promotes neural plate folding and preserves tissue length. Scale bars are 50 μm in (**b**) and 20 μm in (**c**).

shape of cells at the hinges may be driven by passive mechanical responses instead of planar polarized junctional shrinking. Notably, while our mosaic RNA and MO injections suggest a role of intrinsic forces in epithelial folding, similar to previous studies[44,50,51], they do not exclude the role of global forces derived from the surrounding mesodermal tissues in NT closure. These observations highlight similar

roles of mechanical forces in the regulation of cell shape and epithelial tissue dynamics in other systems, especially in *Drosophila* ventral furrow formation[30,52–55].

Early studies argued that isotropic apical surface contraction should cause shortening of the neural tube, contrasting the observed elongation of the anteroposterior body axis[4]. We propose that the

passive reaction of non-constricting cells to the tensile forces of their constricting neighbors causes apical domain expansion and alignment along the AP axis (Fig. 7f). Thus, the presence of these two cell morphologies allows the neural plate to fold at the hinges while maintaining its overall length. We suggest this mechanism as an alternative to other models of NTC[3–6,39]. According to the Takeichi model, NTC in the chick embryo involves PCP-dependent anisotropic shrinking of the cell junctions that are perpendicular to the AP axis[10]. In our model, the force-generating cells are mechanically isotropic, but the forces they produce expand apical domain size and aspect ratio of the adjacent cells. Although the outcome of both models is the same, we expect that the activation of contractility in a selected cell population causes the passive elongation response from their neighbors. Consistent with our model, neural tissue deformations are more significant after laser cutting of the cell junctions that are along the AP axis as compared to the ones perpendicular to the axis[48]. Currently, lack of more direct measurement of tension by laser ablation is limiting the significance of our findings. To distinguish the two models, further studies are needed to compare the subcellular localization of various mechanosensitive proteins and contractility markers.

The morphogenetic behaviors that we described in the neural plate (and, presumably, the underlying physical forces) require intact PCP signaling because they were inhibited by the depletion of Vangl2. This finding supports earlier studies connecting PCP proteins to apical constriction[24,56]. Given the ubiquitous presence of long-range PCP, an important question is whether PCP signaling acts by affecting local or global forces, such as those derived from the adjacent mesoderm. We find that mosaic injections of *vangl2* MO that were targeted exclusively to the ectoderm increase apical domain size in a cell-autonomous manner but minimally affect overall embryo length. These observations suggest that Vangl2 regulates apical domain heterogeneity by acting on local forces that are intrinsic to the manipulated neuroepithelial cells or their neighbors. Nevertheless, the results do not exclude the alternative possibility of PCP proteins active in another tissue (e.g., mesoderm) to modulate a different mechanical force during the same morphogenetic event. Whereas our work focuses on the forces originating in the superficial neural plate, it would be of interest to determine the effect of mesoderm-specific Vangl2 knockdown on the apical domain heterogeneity and bending of the NP.

We propose that the observed changes in cell aspect ratios in Vangl2-depleted neural plate reflect the loss of proper cell orientation in the epithelial plane, i.e. PCP. Of note, PCP protein depletion had an effect on both anterior and posterior regions of the neural plate, whereas the PCP function is commonly associated with the effect of PCP proteins on convergent extension movements in the mesoderm and the posterior neural plate[7,17,57]. This difference is likely due to the fact that our study separately assessed apical domain size in the cell populations along the hinges, which are similar in the anterior and posterior regions of the neural plate while they significantly differ from the cells in non-hinge areas. Alternatively, the observed differences might reflect the varying degree of PCP protein depletion or the different timepoints chosen for the assessment.

Several possibilities may be considered to explain the origin of apical domain heterogeneity in the neural plate. First, global patterning factors, such as neural tissue-inducing BMP inhibitors[58], should play a role, because apical constriction is not normally observed in non-neural ectoderm and because the position of dorsolateral hinges tightly correlates with the neural plate border. Second, the "salt-and-pepper" pattern of cells with large and small apical domains may form stochastically in response to slight variations of contractile activity. Third, the location of the cells with a particular morphology may be transcriptionally controlled and amplified by lateral inhibition, e.g., by the Notch pathway[59]. Our in situ hybridization revealed "salt-and-pepper" distribution of *lmo7* RNA in gastrula dorsal ectoderm at the

time preceding neural plate folding. This finding suggests the existence of a prepattern in the ectoderm. Additional studies are warranted to explain how the contractility of the neural plate is controlled. Despite the general "rule" of alternation between constricted and non-constricted cells, clusters of contractile cells may form if they are triggered to contract their apical surfaces simultaneously or one after another. In *Drosophila* embryos, the onset of apical constriction in some cells was found to correlate with the timing of this process in neighboring cells[60,61]. A more recent study described "cellular constriction chains" (CCCs) that have been associated with positive feedback regulation during propagation of tensile stresses[34]. Whether the morphological changes observed in our study are subject to mechanical, transcriptional or post-transcriptional regulation, they likely play critical roles in many diverse tissues and processes, including *Drosophila* ventral furrow formation[62] or *Xenopus* blastopore[63].

## Methods

### Xenopus embryos, plasmids, and microinjections

Wild-type *Xenopus laevis* were purchased from Nasco and Xenopus1, maintained and handled following the Guide for the Care and Use of Laboratory Animals of the National Institutes of Health. A protocol for animal use was approved by the Institutional Animal Care and Use Committee at Icahn School of Medicine at Mount Sinai. Sex of animals was not considered in the study design and analysis because the study subjects were sexually indifferent embryos. In vitro fertilization and embryo culture were performed as described[25]. Embryo staging was determined according to Nieuwkoop and Faber[64]. For microinjections, embryos were transferred into 3% Ficoll 400 (Pharmacia) in 0.5× Marc's modified Ringer's (MMR) solution (50 mM NaCl, 1 mM KCl, 1 mM CaCl₂, 0.5 mM MgCl2 and 2.5 mM HEPES (pH 7.4)[65].

Plasmids encoding GFP-Lmo7, Flag-Lmo7, and GFP have been previously described[41]. pCS2-myr-tagBFP, -GFP or -RFP were generated by PCR of tagBFP, GFP, or RFP and annealing of chemically synthesized oligonucleotides encoding a myristoylation site. pCS2-3xGFP-mZO1 contains in-frame fusion of three GFP open reading frames and mouse ZO1 cDNA. Details of cloning are available upon request. Capped RNA was synthesized using mMessage mMachine SP6 Transcription kit (Invitrogen) and purified by LiCl precipitation or RNeasy mini kit (Qiagen). *vangl2* MO (5′-CGTTGGCGGATTTGGGTCCCCCGA-3′) was described previously[25]. RNA or MO in 5–10 nl of RNase-free water (Invitrogen) were microinjected into one to four animal blastomeres of 4-16-cell embryos.

### Embryo culture, immunostaining, cryosections, and imaging of Xenopus embryos

For apical domain imaging of neurula embryos, 50 pg of control GFP RNA with or without 5 ng of *vangl2* MO was injected into one dorsal blastomere of four-cell embryos. RNA or MO-injected embryos were cultured in 0.1× MMR until early gastrula or neurula stages. Each experiment included 20–30 embryos per condition. Experiments were repeated at least three times.

For the pigmentation assay during apical constriction in animal pole ectoderm, live embryos were imaged at the required stage using a Leica stereo-microscope equipped with a JENOPTIK GRYPHAX® microscope camera (Jenoptik USA).

Embryos were fixed in MEMFA (100 mM MOPS (pH 7.4), 2 mM EGTA, 1 mM MgSO₄, 3.7% formaldehyde)[66] for 1 h at room temperature. After permeabilization using 0.1% Triton X-100 in PBS for 10 min, embryos or dorsal explants were stained with Alexa Fluor 555-conjugated phalloidin (ThermoFisher Scientific) in PBS containing 1% BSA overnight at 4 °C. The dissected neural plate was mounted on a slide glass with two coverglass spacers (0.13–0.17 mm) to minimize damage to the morphology.

For cryosectioning of the NP, phalloidin-stained embryos were transferred into PBS containing 10% sucrose and 7.5% gelatin. Cross-sections were generated using Leica CM3000 Cryostat.

Images of phalloidin-stained embryos were captured using the LSM980 confocal microscope (Zen (blue edition) Ver 3.7.97.04000, Zeiss) with a 20× dry objective with NA = 0.8 and WD = 0.55 mm (Zeiss), or BC43 spinning disk confocal microscope (Fusion Ver 2, Andor, Oxford Instruments) using a 20× dry objective with NA = 0.8 and WD = 0.8 mm. At least two independent experiments have been done with 20 embryos in each experimental group. After maximum intensity projection, images were converted to tif files using ImageJ2 Ver 2.9.0/1.54 g for analysis and quantification.

For time-lapse imaging, embryos injected with GFP-Lmo7 RNA or a lineage tracer RNA (myrBFP, myrRFP, GFP-CAAX, or 3GFP-ZO1) (100 pg) were cultured until gastrula. Embryos were mounted in 1% low melting temperature agarose (Lonza) on a slideglass attached with a silicone isolator (Grace Bio-labs) or on a glass-bottom dish (Cellvis). Time-lapse imaging was carried out using the AxioZoomV16 fluorescence stereomicroscope (Zeiss) equipped with the AxioCam 506 mono CCD camera (Zeiss) or the LSM880 confocal microscope. Images were taken every 6–10 min for the period of 1.5–4 h. Apically constricted (AC) or expanded (AE) cells were defined as the ones in which the apical domain has been reduced or expanded, respectively, by more than 20%.

### In situ hybridization
In situ hybridization with anti-sense and sense *lmo7* probes was performed as previously described[41]. Embryo images were captured using a Leica stereo-microscope equipped with a Jenoptik GRYPHAX® microscope camera (Jenoptik USA).

### Data analysis, segmentation, tracking, apical domain assessment
Grayscale images of the cell outline marker were segmented using the Python package Cellpose v2.0.5[67]. Each frame was first segmented using the pre-trained model *cyto2* and manually corrected as needed. Cell morphology was quantified for each cell interior mask after post-processing steps to ensure tissue confluence. The cell aspect ratio was measured as the ratio between the lengths of the long and short axes of the ellipse with the same normalized second central moments as the cell[68]. The angle between the long axis and a reference direction defined the cell orientation. Segmented cells were tracked across timepoints using the Python package Bayesian Tracker (btrack v0.4.5)[69], where the cell centroid, area, major and minor axes lengths, perimeter, solidity, and higher central moments (up to 3rd order) were used as features in Bayesian Tracker's probabilistic model.

For the NP segmentation, the NP area in stage 14–15 embryos was defined by the brighter phalloidin staining as compared to the surrounding epidermis. Dorsolateral hinge regions were defined as 4-5-cell-wide corridors at the border of the NP. The remaining NP was defined as "non-hinge" region.

### Analysis of junctional and cytoplasmic fluorescent intensities
The morphology and fluorescent intensity statistics of individual cell junctions were quantified using a custom pipeline. After segmentation and tracking, each junction was first identified as a vertex or edge of the cell outline network and mean protein intensities were measured using dilated masks of the pixel or set of pixels that defined the junction. Cytoplasmic fluorescent intensities were similarly measured using eroded cell interior masks.

### Statistics and reproducibility
Histograms, dot plots and rose plots for experimental data were generated using the GraphPad Prism 10 (Ver 10.1.0) and Rose Diagram

Creator. For cell orientation, the AP axis was set at 0°. Individual experiments were repeated at least three times. The coefficient of variation (CV) and s. d. were calculated using the GraphPad Prism 10 and used to measure the variability in individual sample groups. The Kolmogorov-Smirnov test was used to determine statistical significance of difference between two groups with non-normal distribution. One-way ANOVA Kruskal-Wallis test was used to determine statistical significance of the difference among more than three groups. The Freeman-Halton extension of Fisher's exact test was used to determine statistical significance of the difference calculated from the contingency tables larger than $2 \times 2$.

### 2D vertex model
Each cell in the 2D vertex model is a polygon. The evolution of the system follows the friction-dominated equation of motion for the polygon vertices

$$\eta \dot{\mathbf{R}}_i = - \nabla_{\mathbf{R}_i} E, \tag{2}$$

where $\mathbf{R}_i$ is the position of vertex $i$, $\eta$ is the viscosity of the vertices, $\nabla_{\mathbf{R}_i}$ indicates the gradient relative to the position of the $i$-th vertex, and $E$ is the (dimensionful) energy of the tissue. We use the usual area-and-perimeter-elasticity energy function[27]:

$$E = \sum_c^N \left[ K_A \left( A^{(c)} - A_0^{(c)} \right)^2 + K_P \left( P^{(c)} - P_0^{(c)} \right)^2 \right]. \tag{3}$$

Here, $A^{(c)}$ and $P^{(c)}$ are the area and perimeter, respectively, of cell $c$, $A_0^{(c)}$ and $P_0^{(c)}$ are respectively its target area and perimeter, whereas $K_A$ and $K_P$ are respectively the area and perimeter elasticity moduli. To recast the equation in dimensionless form, we divide this energy by $K_A A_0^2$, where we assume that the target area of all cells is identical, $A_0^{(c)} = A_0$. This results in the main text Equation, with the definitions $e = E/(K_A A_0^2), a^{(c)} = A^{(c)}/A_0, k_p = K_P/(K_A A_0), p^{(c)} = P^{(c)}/\sqrt{A_0}$, and, $p_0^{(c)} = P^{(c)}/\sqrt{A_0}$. The equation of motions then becomes

$$\dot{\mathbf{r}}_i = - \nabla_{\mathbf{r}_i} e, \tag{4}$$

where $\mathbf{r}_i = \mathbf{R}_i/\sqrt{A_0}$ is the dimensionless position of the $i$-th vertex, the overdot indicates the derivative with respect to the dimensionless time (measured in units of $\eta/(K_A A_0)$), and the gradient is now with respect to the dimensionless position of vertex $i$.

The initial condition of our model is a regular hexagonal lattice with $N'_x$ and $N'_y$ cells along the $x$ and $y$ axis, respectively. In the model neural plate region, every second row contains an extra cell along the $y$ axis, so that the region is symmetrical relative to the AP axis. The outermost cells of the entire model tissue cannot change shape, enforcing fixed boundary conditions. The resulting equations of motion are solved using an explicit Euler scheme, with time step $\Delta t = 0.0001$, and model tissues are shown at time $t = 2000$ (in the case of the $N'_x = N'_y = 180$ tissue shown in Fig. S5 e, f, we used $\Delta t = 0.001$ due to the computational intensity of simulating a model tissue of that size, and the tissue is shown at $t = 20000$ as the greater number of cells means it takes longer for it to relax). Note that our model does not allow for T1 transitions and the model tissue is therefore always a hexagonal lattice in terms of cell neighbor relations.

### Cell elongation in vertex models
In both 2D and 3D, we define the direction of elongation of a cell as the eigenvector corresponding to the highest eigenvalue of the cell's gyration tensor.

$$S^{(c)} = \frac{1}{n^{(c)}} \sum_{i=1}^{n^{(c)}} \left( \mathbf{r}^{(i)} - \mathbf{r}_{\text{center}}^{(c)} \right) \bigotimes \left( \mathbf{r}^{(i)} - \mathbf{r}_{\text{center}}^{(c)} \right). \tag{5}$$

Here, the sum is over the $n^{(c)}$ vertices corresponding to cell $c$ (or its apical side in the 3D case), $\mathbf{r}_{center}^{(c)}$ is the average position of all vertices corresponding to that cell (or cell side), and $\otimes$ indicates the outer product. To calculate the elongation of a cell, we use

$$\kappa^{(c)} = \frac{g_1^{(c)} - g_2^{(c)}}{g_1^{(c)} + g_2^{(c)}}, \qquad (6)$$

where $g_1^{(c)}$ and $g_2^{(c)}$ and respectively the first and second largest eigenvalues of $S^{(c)}$. Lastly, to determine the length and width of the model neural plate region for Fig. 3d, we first select several centrally located cells at each of the outer edges of the model neural plate region. After tissue relaxation, we then measure the distance between the mean centers of relevant pairs of these cell groups.

### 3D vertex model of furrow formation

In the 3D model, cells are represented as polyhedrons with near-constant volume $V_c$ arranged in a mono-layered sheet. The apical and basal sides of the cells are polygons, whereas the lateral sides are quadrilaterals; we use a model based on surface-tensions in cell sides[70-73], expanded to also include an apical-perimeter-elasticity term[74]. The energy function, therefore, reads

$$E = \sum_c^N \left[ \Gamma_a A_a^{(c)} + \Gamma_b A_b^{(c)} + \frac{1}{2}\Gamma_l A_l^{(c)} + K_P \left( P_a^{(c)} - P_0^{(c)} \right)^2 + K_V \left( V^{(c)} - V_c \right)^2 \right], \qquad (7)$$

where the sum runs over all $N_c$ cells, $A_a^{(c)}$, $A_b^{(c)}$, and $A_l^{(c)}$ are the apical, basal, and lateral areas of the $i$-th cell, whereas $\Gamma_a$, $\Gamma_b$ and $\Gamma_l$ are the corresponding surface tensions and the factor $1/2$ arises because each lateral side is shared between two cells. $P_a^{(c)}$ and $P_0^{(c)}$ are the current and target apical perimeter of the $i$-th cell, with $K_P$ being the perimeter elasticity modulus. $V^{(c)}$ is the volume of the $i$-th cell, $V_c$ is the target cell volume, and $K_V$ is the bulk modulus.

To non-dimensionalize the model, we divide it by $\Gamma_l V_c^{2/3}$, so that the dimensionless energy now reads

$$\sum_c^N \left[ \alpha a_a^{(c)} + \beta a_b^{(c)} + \frac{1}{2} a_l^{(c)} + k_p \left( p_a^{(c)} - p_0^{(c)} \right)^2 + k_V \left( v^{(c)} - 1 \right)^2 \right]. \qquad (8)$$

Here, $\alpha = \Gamma_a/\Gamma_l$ and $\beta = \Gamma_b/\Gamma_l$ are the dimensionless apical and basal surface tensions, respectively; $a_a^{(c)} = A_a^{(c)}/V_c^{2/3}$, $a_b^{(c)} = A_b^{(c)}/V_c^{2/3}$, and $a_l^{(c)} = A_l^{(c)}/V_c^{2/3}$ are the apical, basal, and lateral dimensionless areas of the $c$-th cell, respectively; $p_a^{(c)} = P_a^{(c)}/V_c^{1/3}$ and $p_0^{(c)} = P_0^{(c)}/V_c^{1/3}$ are the dimensionless actual and target apical perimeter of the $c$-th cell, and $v^{(i)} = V^{(c)}/V_c$ is the dimensionless volume of that cell. The dimensionless perimeter elasticity modulus is given by $k_p = K_P/\Gamma_l$, whereas the dimensionless bulk modulus is given by $k_V = K_V V_c^{4/3}/\Gamma_l$. We set $\alpha = \beta = 0.5$, corresponding to a cuboidal tissue, $k_p = 1$, and $k_V = 100$. The target apical perimeter is set to equal the equilibrium apical perimeter of a cell without perimeter elasticity in a flat hexagonal honeycomb tissue, $2 \cdot 6^{1/3}(\alpha + \beta)^{-1/3}$. To model the constricting cells, we again set their target apical perimeter to equal to 10% of the target perimeter of non-constricting cells. We assume overdamped equations of motion, with dimensionless vertex positions now measured in units of $V_c^{1/3}$ and dimensionless time in units of $\eta/\Gamma_l$. The resulting equations of motion are solved using an explicit Euler scheme, with time step $\Delta t = 0.0001$; model tissues are shown at $t = 5000$.

The simulations again start with a regular hexagonal lattice. We set boundary conditions along the AP axis to be periodic, as the AP length of the hinge is much larger than its width. Boundary conditions perpendicular to AP are fixed. The simulated patch has 20 cells along the AP axis, and 40 perpendicular to it. As before, the outermost cells perpendicular to AP cannot change shape. Moreover, T1 transitions are again not allowed.

### Reporting summary

Further information on research design is available in the Nature Portfolio Reporting Summary linked to this article.

### Data availability

All data supporting the findings of this study are available within the paper, its supplementary information and source data file provided with this paper. Source data are provided with this paper.

### Code availability

The custom code used in this study to analyze protein localization patterns and tissue structure is available at https://github.com/kaszalab/xenopus-protein-localization.

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

## Acknowledgements
We thank Stas Shvartsman, Rastko Sknepnek, Olga Ossipova, Kyeongmi Kim, and the current members of the Sokol laboratory for valuable discussions and Vladimir Gelfand and Mark Corkins for comments on the manuscript. We also wish to thank Matej Krajnc for providing the initial version of the vertex model codes and Damian Dalle Nogare for providing the code for segmentation, tracking, and apical domain assessment at the initial stage of this study. We acknowledge the help from the ISMMS Microscopy Core facility. This paper is dedicated to the memory of Olga Ossipova, whose unpublished observations and original ideas served as a basis for our study. This research was supported by the NIH grants R35GM122492 and R01NS100759 to S.Y.S., and R35GM138380 to K.E.K. K.E.K. holds an NSF CAREER Award, Packard Fellowship, and Sloan Research Fellowship in Physics. J.R. acknowledges support from the UK EPSRC (Award EP/W023849/1).

## Author contributions
M.M., J.R., and S.Y.S. initiated, designed the experiments and developed the project. M.M. performed all the experiments, analyzed the data and prepared the figures. J.R. developed and tested the vertex model. S.O. and K.E.K. developed the code for image segmentation and quantification. M.M. and S.Y.S. wrote the manuscript. All authors contributed to data interpretation and manuscript writing. All authors approved the final manuscript.

## Competing interests
All authors declare no competing interests.
