## [Peer Review File · Nature Communications]

Mechanical control of neural plate folding by apical domain alterationREVIEWER COMMENTS

Reviewer #1 (Remarks to the Author):

This paper using *Xenopus laevis* embryo focuses on the heterogeneity of apical cell shape, especially of the hinge region of neural plate which may play a pivotal role in neural plate folding. A new finding is that alternating cells with highly constricted apical area as well as the cells elongated along the AP axis are observed in the hinge region. Based on that, the authors claim that mechanical control and PCP signaling contribute to the morphological event. Although the data are solid and convincing, there are several weak points to demonstrate the role of physical force in this phenomenon.

Major points

It is suspected that a “tug-of-war” relationship of cells exists in the hinge region; constricting cells generate pulling force by which neighboring cells are pulled along the AP axis. If this is true, it is worthy testing whether the elongating cells are indeed under a certain tension, which could be demonstrated, for example, by a laser ablation. As further attempts to show the presence of physical interaction of cells, weakening cell-cell adhesion may be a useful experiment as the interaction is thought to be relied on cell adhesion. This may be difficult to apply for neural plate but the model using animal side ectoderm (Fig 5) should be a doable system.

Related to above comments, the apical cell shape heterogeneity may come from a biased distribution of F-actin (less abundant on cell membranes along the AP axis) which may contribute to AP elongation of cells. Therefore, it is important to quantitate the F-actin intensity of the Phalloidin staining in Fig. 2. The biased F-actin may be the cause for the overall tissue morphogenesis. It is also important to show the F-actin localization is affected in *vangl2* morphants.

Another question is, does overall tension along the AP axis due to the elongation of whole embryo rather than the local pulling force by constricting cells affect the heterogeneity of apical cell shape? This question might be difficult to address as many of elongation-defect embryos are caused by mutants of PCP genes. However, the external forces are something to be considered as the possible source of force regulating the heterogeneity of cell shape in the hinge region.

Finally, given a tug-of-war relationship is essential, the important question would be how the constricting and elongating cells be generated? Is there prepattern exist or do elongating cells emerge in a passive manner to balance the force along the AP axis generated by stochastically appearing constricting cells? As the authors partly addressed this question in Fig.6 using an ectoderm model, more direct investigations focusing on force would be necessary.

Minor points

Fig. 2D, the variation of cell aspect ratio is not evident compared to the data of Fig. 1F and thus the change in the ratio in *vangl2KD* embryo does not seem to be so significant. How is this explained? Embryo's batch effect?

In Fig. 2 C, D, “apical domain size” and “cell aspect ratio” on the top, could be deleted and shown as Fig.1 E, F labeled as apical domain size (μm^2) and cell aspect ratio, respectively.

In Fig. 3A, the number of shrinking cells set for the simulation comparable to the actually observed shrinking cells? If the number adjusted to more probable number, what would be the result of the simulation?

In Fig. 3, does the boundary condition of the initial shape of the virtual tissue affect the simulation? In other words, what would happen if the simulation is run on a square sheet of virtual tissue?

Reviewer #2 (Remarks to the Author):

As neural tube closure defects are amongst the most frequent human birth defects, there is a significant interest in understanding the genetic as well as the mechanical mechanisms driving this complex process. The manuscript by Matsuda et al. "Mechanical control of neural plate folding by apical domain alteration" aims to add to our understanding by analyzing cellular mechanisms contributing to vertebrate neural tube closure. As a model system the authors used the *Xenopus* neural plate, where they observed a mixture of apically constricting and extending cells. This variable morphology of the apical domains was most apparent in the hinge areas of the neural plate. As morpholino-mediated knockdown of the PCP protein *Vangl2* prevented apical domain heterogeneity, the authors suggest that this phenomenon is likely controlled by PCP signaling. To test if the apical contraction of a limited number of cells is sufficient to mechanically cause elongation of neighboring cells, the authors used vertex model simulations. They predict that the geometry of the tissue depends on the frequency and distribution of the apically constricting cells and they propose a model whereby constriction of a limited number of cells is sufficient to cause cell elongation and orientation in the neural plate. To test this hypothesis *in vivo*, the authors induced ectopic apical constriction in *Xenopus* gastrula ectoderm by overexpressing *Lmo7* or *Shroom3*. Consistent with their model they observed the formation of a pigmented furrow.

Considering recent publications concerning the mechanical control of neural tube closure, this - as the authors state - minimal model, seems surprisingly simple; especially, since Baldwin et al. (eLife, 2022) observed distinct patterns of apical constriction behavior in the anterior versus posterior closing neural tube. Furthermore, Baldwin et al showed differential apical constriction phenotypes between regions of the neural ectoderm in mosaic *shroom3* crispants. Thus, a more detailed discussion of this contradictory findings would be advisable.

The overexpression studies presented here are intriguing and consistent with the proposed model. However, they were performed at late gastrula stages and not at neural tube closure stages. Furthermore, open questions remain, for example, why does uniform overexpression of *Lmo7* or *Shroom3* only lead to apical constriction of few cells, that sometimes seem also to appear in clusters?

In the following I will list few points where I think additional information is required:

1. The authors state that PCP signaling functions upstream of the apical domain heterogeneity. As evidence they present *Vangl2* loss of function experiments. However, the observed *Vangl2* morphant phenotype could also simply be caused by a delay in neural tube closure, which is not uncommon for knockdown of PCP players with a function in neural tube closure. Can the authors rule out that the lack of apical constriction is not simply the result of delayed closure? Do these embryos show apical domain heterogeneity at later stages on the

injected side?

2. It is difficult to distinguish which areas of the neural plate have been analyzed. For example, Fig. 1A shows the medial hinge area, while Fig. 1D analyzes the dorsolateral hinge area. What do we see in Fig. 1 E, both? What was analyzed in Fig. 2? The hinge area, the neural plate or both? This information would also be important to compare these data to previous publications.
3. How did the author determine the medial hinge area, what landmarks were they using? And how did they determine constricting versus elongated cells in fixed neural plate tissue (e.g. Fig. 1B)?
4. Why is there such a variance between the frequency in apical domain size (cell aspect ratio) in the ectoderm of control cells in Fig. 1E/F and Fig. 7D/E?
5. It would also be helpful to comment why cells of the ectoderm of stage 11 embryos were analyzed in Fig. E/F. This becomes clear when the ectopic expression data is presented, but is confusing in Fig. 1. Alternatively, these data can be moved to the supplement.
6. Why are the cells in Fig. 6F/F' marked with stars?
7. A more detailed description of the model in Fig. 7F would be helpful.

Reviewer #3 (Remarks to the Author):

This paper points out the changes in cell shape that occur in the posterior neural plate during the rolling up, and the convergence and extension of the neural tube. While Shroom3 mediated apical constriction occurs as part of the rolling up, cells adjacent to constricting cells are shown here to undergo apical expansion, and to be elongated in the anterior posterior axis. This behavior accommodates the elongation of the neural plate while allowing it to converge medolaterally.

Modeling shows that in principle this effect can be caused in a rectangular field by apical constriction of some cells, with the overall forces of the rectangle leading to apical expansion of the apices of cells that are not actively constricting, with stretching of cells in the long orientation.

In addition, to modeling the cell behaviors, Lmo7 is used as a mediator of apical constriction precociously in the blastula stage, and is shown to induce similar flattening of cells, thus illustrating that this model could apply to the normal AP elongation of non-apically-constricting cells. The model and experiment support the authors contention that this mixture of AP elongating cells with apically constricting cells may be sufficient to allow elongation of the neural plate during neural plate closure. However the argument does not include the likely contribution of other forces in neural tube closure, nor does it provide a satisfactory explanation for the requirement for Planar Cell Polarity signaling. For example, the flattening of cells has been discussed before, e.g. example by Hardin and Keller, who ascribed this to the stiffness of adjacent tissues.

The model does not seem to apply in the anterior neural plate, where there is less PCP dependent neural tube narrowing, and less extension. So far as I see, particularly from the movies in Butler and Wallingford also much less stretching of non constricting cells, yet the neural tube rolls up effectively. This behavior provides a good comparison and should be

more effectively considered and contrasted as a different set of behaviors. I note that recent work from Baldwin and Wallingford has used careful analysis of apical expansion and constriction, and that work also needs to be compared and contrasted more thoroughly in the manuscript.

Since the authors essentially dismiss a significant role of the mediolateral intercalation of the cells in the normal neural plate closure, I think it is important to review the data from Butler and their own data to see how much quantitatively it contributes to narrowing of the posterior neural plate. I don't see that this lack of importance has been presented elsewhere in the literature.

Perhaps more importantly, the authors need consideration of the other forces that drive the narrowing of the neural plate, for example the intercalation and migration of deep cells, (reviewed by Davidson and Keller in 1999). These deep cells rearrange dramatically. While not readily visible, they are also influenced by PCP, and have previously been assumed to exert forces to close the neural tube. so their conclusion "We propose that neural folding relies on PCP-dependent transduction of mechanical signals between neuroepithelial cells. " may well be incorrect, since these forces may largely derive from the deep neural plate cells acting mechanically on the overlying epithelium.

In addition, there is proposed to be medial force from the medial migration of the neural folds and epidermis.

It is very difficult to tease apart the contributions of the different forces. Ideally, if the authors wish to argue that the movements and shapes they see are sufficient to roll up the neural tube, then this ought to be feasible in explanted tissue, perhaps plated on a pliable sheet, to study the effects of PCP more directly on the neuroepithelial cells.

Despite the drawbacks, the authors make a useful contribution, but as it stands it is incomplete, both in terms of placing it in the context of the current literature and in the level of understanding of how the cells become elongated or compressed. The role of PCP is not shown to be a direct one on the epithelium, and so it is important to put that into context also. Of course it is understandable that the authors want to emphasize their own contributions, but the failure to put their work in proper context does appear as ungenerous or selective. For example: "We conclude that the neural plate acquires considerable cell heterogeneity by the onset of folding, a result supported by another study 7. " is simply the wrong presentation. Chronologically it is hard to see how an older study supports a newer study, and the authors need to turn the sentence around

RESPONSES TO THE REVIEWERS' COMMENTS

We thank all three reviewers for their thorough and constructive comments that helped us improve the manuscript. Our responses to the reviewers' comments and the major changes in the revised text are indicated by altered font color.

Reviewer #1

*This paper using *Xenopus laevis* embryo focuses on the heterogeneity of apical cell shape, especially of the hinge region of neural plate which may play a pivotal role in neural plate folding. A new finding is that alternating cells with highly constricted apical area as well as the cells elongated along the AP axis are observed in the hinge region. Based on that, the authors claim that mechanical control and PCP signaling contribute to the morphological event. Although the data are solid and convincing, there are several weak points to demonstrate the role of physical force in this phenomenon.*

Major points

It is suspected that a “tug-of-war” relationship of cells exists in the hinge region; constricting cells generate pulling force by which neighboring cells are pulled along the AP axis. If this is true, it is worthy testing whether the elongating cells are indeed under a certain tension, which could be demonstrated, for example, by a laser ablation.

We followed the suggestion of the referee and found that the laser ablation experiment and junctional recoil measurement at subcellular resolution are technically challenging to carry out in the neural plate due to dynamic cell rearrangements. Nevertheless, previous studies support the view that the neural plate is under anisotropic tension. When a cut was placed orthogonally to the anteroposterior (AP) axis, we observed that the wound expanded faster than when the incision was parallel to the AP axis ((Mancini et al., 2021) and data not shown). Importantly, another study using laser ablations in the *Xenopus* neural plate reached a similar conclusion (Hirano et al., 2022). These observations are now discussed in the revised text (p. 15) to further support our hypothesis that the elongating junctions along the AP body axis are under higher tension.

As further attempts to show the presence of physical interaction of cells, weakening cell-cell adhesion may be a useful experiment as the interaction is thought to be relied on cell adhesion. This may be difficult to apply for neural plate but the model using animal side ectoderm (Fig 5) should be a doable system.

We followed the suggestion from the reviewer and cultured the neural explant in the presence of different amounts of EDTA (up to 1 mM) to reduce cadherin-mediated adhesion. This treatment did not decrease but slightly increased apical domain heterogeneity (data not shown). We feel that this result does not exclude the role of cell adhesion in force transmission between cells, because low levels of cadherins required for tissue integrity are still present in the treated embryos. We therefore decided not to include this experiment in our manuscript.

Related to above comments, the apical cell shape heterogeneity may come from a biased distribution of F-actin (less abundant on cell membranes along the AP axis) which may contribute to AP elongation of cells. Therefore, it is important to quantitate the F-actin intensity of the Phalloidin staining in Fig. 2. The biased F-actin may be the cause for the overall tissue morphogenesis.

It is also important to show the F-actin localization is affected in vangl2 morphants.

We acknowledge that the unequal distribution of F-actin is likely to play a role in neural plate folding. The quantification of F-actin in wild-type and morphant neural plates has been included in new Fig. S2. In wild-type neural plates, mediolaterally oriented junctions (that are perpendicular to the AP axis) were shorter and had stronger phalloidin staining, consistent with previously described enrichment of F-actin at shorter junctions (Baldwin et al., 2023; Butler and Wallingford, 2018). In *vangl2* morphants, the overall intensity of junctional F-actin was lower than in the uninjected cells regardless of junction length (Fig. S2).

Another question is, does overall tension along the AP axis due to the elongation of whole embryo rather than the local pulling force by constricting cells affect the heterogeneity of apical cell shape? This question might be difficult to address as many of elongation-defect embryos are caused by mutants of PCP genes. However, the external forces are something to be considered as the possible source of force regulating the heterogeneity of cell shape in the hinge region.

The referee asks whether the observed apical domain changes depend on local or global forces derived from the adjacent tissues. Mosaic depletion of Vangl2 in the ectoderm caused clear cell-autonomous effects on apical domain size (new Fig. S3). In these mosaic injections, Vangl2 MO was properly targeted to the ectoderm, largely absent from the mesoderm (new Fig. S4A, B), and minimally affected the length of the entire embryo (new Fig. S4C). These observations argue that the effect of Vangl2 depletion on apical domain size is unlikely to be mediated by altered global forces. Rather, our data suggest that PCP signaling regulates apical domain heterogeneity via local forces that are intrinsic to the manipulated cells or their neighbors. This issue has been addressed in the revised Discussion. Please also see our related responses to Reviewer 3.

Finally, given a tug-of-war relationship is essential, the important question would be how the constricting and elongating cells be generated? Is there prepattern exist or do elongating cells emerge in a passive manner to balance the force along the AP axis generated by stochastically appearing constricting cells? As the authors partly addressed this question in Fig.6 using an ectoderm model, more direct investigations focusing on force would be necessary.

We agree with the referee that the question how the constricting and elongating cells are generated in the neural plate is intriguing. So far, our findings are consistent with the model, in which stochastically appearing apically constricting cells are passively balanced by elongating cells. To address how apically constricting cells are first selected, we

examined spatial distribution of transcripts for *Lmo7*, a regulator of apical constriction in the ectoderm (Matsuda et al., 2022). *In situ* hybridization revealed “salt-and-pepper” appearance of *lmo7* RNA in gastrula dorsal ectoderm at the time preceding the observed apical domain regulation in the neural plate (new Fig. S9). This finding suggests the existence of molecular differences or prepatterning in the ectoderm, but more work is needed to uncover its origin. We feel that the detailed mechanistic analysis of this question is beyond the scope of the current paper and should be a subject of future studies.

Minor points

Fig. 2D, the variation of cell aspect ratio is not evident compared to the data of Fig. 1F and thus the change in the ratio in vangl2KD embryo does not seem to be so significant. How is this explained? Embryo's batch effect?

In Fig. 1F, cell aspect ratios in the hinge and non-hinge regions were independently computed in the same embryos. In Fig. 2D, the hinge and non-hinge regions were quantified together because it is hard to distinguish dorsolateral hinge regions in *Vangl2* morphants. Since the hinge cell population is relatively small compared to the non-hinge cell number, this computation reduced the variability of cell aspect ratios in Fig. 2D. We apologize for the confusion and have made this point clear in the legends.

In Fig. 2 C, D, “apical domain size” and “cell aspect ratio” on the top, could be deleted and shown as Fig.1 E, F labeled as apical domain size (um²) and cell aspect ratio, respectively.

We have modified Fig. 2 as requested.

In Fig. 3A, the number of shrinking cells set for the simulation comparable to the actually observed shrinking cells? If the number adjusted to more probable number, what would be the result of the simulation?

In Fig. S6, we show an example of a more realistic simulation which separates the model posterior neural plate into a hinge region in which cells constrict with a 50% probability, whereas those in the rest of the model neural plate constrict with a 20% probability. This results in non-constricting cells in the hinges having a clear alignment along the AP axis, whereas those in the rest of the model neural plate do not have an obvious alignment with the AP axis.

In Fig. 3, does the boundary condition of the initial shape of the virtual tissue affect the simulation? In other words, what would happen if the simulation is run on a square sheet of virtual tissue?

In the revised manuscript, we clarify that the results do depend on the shape of the external region (p. 9 and new Fig. S5). Specifically, placing the model neural plate at the center of a 100 cell by 100 cell square virtual tissue causes the alignment of non-

constricting cells with the AP axis to become more pronounced.

Reviewer #2:

*As neural tube closure defects are amongst the most frequent human birth defects, there is a significant interest in understanding the genetic as well as the mechanical mechanisms driving this complex process. The manuscript by Matsuda et al. "Mechanical control of neural plate folding by apical domain alteration" aims to add to our understanding by analyzing cellular mechanisms contributing to vertebrate neural tube closure. As a model system the authors used the *Xenopus* neural plate, where they observed a mixture of apically constricting and extending cells. This variable morphology of the apical domains was most apparent in the hinge areas of the neural plate. As morpholino-mediated knockdown of the PCP protein *Vangl2* prevented apical domain heterogeneity, the authors suggest that this phenomenon is likely controlled by PCP signaling. To test if the apical contraction of a limited number of cells is sufficient to mechanically cause elongation of neighboring cells, the authors used vertex model simulations. They predict that the geometry of the tissue depends on the frequency and distribution of the apically constricting cells and they propose a model whereby constriction of a limited number of cells is sufficient to cause cell elongation and orientation in the neural plate. To test this hypothesis in vivo, the authors induced ectopic apical constriction in *Xenopus gastrula* ectoderm by overexpressing *Lmo7* or *Shroom3*. Consistent with their model they observed the formation of a pigmented furrow.*

*Considering recent publications concerning the mechanical control of neural tube closure, this - as the authors state - minimal model, seems surprisingly simple; especially, since Baldwin et al. observed distinct patterns of apical constriction behavior in the anterior versus posterior closing neural tube. Furthermore, Baldwin et al showed differential apical constriction phenotypes between regions of the neural ectoderm in mosaic *shroom3* crispants. Thus, a more detailed discussion of this contradictory findings would be advisable.*

The revised Discussion compares our study with other recent papers that described significant differences in cell behavior between the anterior and posterior neural plate based on time-lapse imaging of live embryos. In general, cell intercalation behavior was followed by apical constriction in the posterior NT, whereas apical constriction was continuously observed in the anterior NP (Baldwin et al., 2022; Christodoulou and Skourides, 2022a; Christodoulou and Skourides, 2022b). We report an increase in apical domain heterogeneity and junctional F-actin enrichment in both the anterior and the posterior NP in phalloidin-stained embryos (Figs. 2, S1, S2). In the posterior NP, we find that the cell junctions along the ML axis are shorter as compared to the junctions along the AP axis. The main difference of our work from the above papers is that we separately assess the constricting cell population at the hinge and non-hinge regions of the NP. In the hinge regions of the anterior NP, we observed a similar reverse correlation between junction length and F-actin enrichment (data not shown). It is important to mention that

cells elongated parallel to the orientation of hinges. Whereas Baldwin et al. showed differential phenotypes of *shroom3* crispants in the anterior vs posterior NP, they did not independently evaluate the hinge and non-hinge cell populations. Please also see our response to Reviewer 3.

The overexpression studies presented here are intriguing and consistent with the proposed model. However, they were performed at late gastrula stages and not at neural tube closures stages. Furthermore, open questions remain, for example, why does uniform overexpression of Lmo7 or Shroom3 only lead to apical constriction of few cells, that sometimes seem also to appear in clusters?

The folding neural plate is under control of many signaling pathways which affect cell lineages and physical forces required for morphogenesis. Gastrula ectoderm is a less complex model in which the contribution of signaling factors to apical domain heterogeneity and epithelial bending can be largely excluded. At the same time, the frequency of spontaneous apical constriction ('noise') in this population is very low. Similar experiments would be harder to interpret if they are carried out in the neural plate.

The reviewer also brings up an interesting question why apical constriction is observed in clusters of cells in addition to individually constricted cells. In the ectoderm overexpressing Lmo7 or Shroom3, we predominantly observe the expansion or elongation of cells adjacent to constricting cell(s). However, contractile cell clusters of Lmo7 or Shroom3 form when they are surrounded by less contractile wild-type cells. Along with the progress of NT closure, the number of contractile cells may increase in the neural plate, which may be ultimately responsible for the formation of apically constricted cell clusters. We note that in *Drosophila* ventral furrow, the onset of apical constriction in some cells was found to correlate with the timing of this process in neighboring cells (Sweeton et al., 1991; Xie and Martin, 2015). A more recent study described 'cellular constriction chains' (CCCs) that have been associated with positive feedback regulation during propagation of tensile stresses (Holcomb et al., 2021). We propose that the mode of 'collective apical constriction' may be different in diverse experimental models depending on contractility parameters. Further *in vivo* or *in silico* studies will test this hypothesis.

In the following I will list few points where I think additional information is required:

1. The authors state that PCP signaling functions upstream of the apical domain heterogeneity. As evidence they present Vangl2 loss of function experiments. However, the observed Vangl2 morphant phenotype could also simply be caused by a delay in neural tube closure, which is not uncommon for knockdown of PCP players with a function in neural tube closure. Can the authors rule out that the lack of apical constriction is not simply the result of delayed closure? Do these embryos show apical domain heterogeneity at later stages on the injected side?

Our main argument against defective NT closure affecting apical constriction is based on the analysis of mosaic morphant cells. As we stated in response to reviewer 1, mosaic

depletion of Vangl2 in the dorsal ectoderm minimally affected embryo body length (Fig. S4C) and overall NT closure in particular in the posterior part of NT (data not shown). However, mosaic Vangl2 knockdown caused obvious cell-autonomous effects on cell shape and apical domain size (Fig. S3). If apical constriction defects are the secondary outcome of defective NT closure, they would not be expected to take place specifically in the Vangl2-depleted cells.

2. It is difficult to distinguish which areas of the neural plate have been analyzed. For example, Fig. 1A shows the medial hinge area, while Fig. 1D analyzes the dorsolateral hinge area. What do we see in Fig. 1 E, both? What was analyzed in Fig. 2? The hinge area, the neural plate or both? This information would also be important to compare these data to previous publications.

In the revised legends related to Fig. 1 and Fig. 2, we included the description of the areas used for analyses in individual panels. In brief, Fig 1D and E include both the medial and dorsolateral hinge areas. The non-hinge areas exclude four-to-five rows of cells from the medial and dorsolateral hinges. The entire neural plate including both hinge and non-hinge areas has been analyzed in Fig 2C and 2D.

3. How did the author determine the medial hinge area, what landmarks were they using?

First, the midline of the whole embryos was determined based on the relative location of the blastopore and the fused regions in the posterior NP (Fig. S1A-B”). Four-to-five rows of cells at the midline were considered the medial hinge area. Morphologically, these cells are the first neuroepithelial cells near the midline that become visibly heterogeneous with respect to their apical domain.

And how did they determine constricting versus elongated cells in fixed neural plate tissue (e.g. Fig. 1B)?

We agree that it is not possible to distinguish “constricting” or “elongating” cells in fixed embryos. In the revised manuscript, we changed the description to “cells with *reduced or expanded* apical domain”, respectively, in comparison to their neighbors.

4. Why is there such a variance between the frequency in apical domain size (cell aspect ratio) in the ectoderm of control cells in Fig. 1E/F and Fig. 7D/E?

We apologize for the confusion. In Fig. 1E/F, stage 11 ectoderm is the wild-type cell population in wild-type embryos, whereas in Fig. 7D/E, Shroom3-positive cells are compared with their adjacent wild-type cells as controls. These wild-type neighbors increase their apical domain size in response to Shroom3-containing cells. We have clarified their description in Fig. 7 legend and changed the name on the graph (‘adjacent wt’ instead of ‘control’).

5. *It would also be helpful to comment why cells of the ectoderm of stage 11 embryos were analyzed in Fig. 7E/F. This becomes clear when the ectopic expression data is presented, but is confusing in Fig. 1. Alternatively, these data can be moved to the supplement.*

As mentioned above, Fig.7E/F shows the directional elongation of wild-type cells that are adjacent to Shroom3-positive constricting cells. In the revised manuscript, we clarified the description of the cell population used as controls.

6. *Why are the cells in Fig. 6F/F' marked with stars?*

Asterisks mark the cells undergoing mitosis during the imaging. This clarification has been missing and is now included in the legend.

7. *A more detailed description of the model in Fig. 7F would be helpful.*

We have expanded our description of the model in the figure legend.

Reviewer #3

This paper points out the changes in cell shape that occur in the posterior neural plate during the rolling up, and the convergence and extension of the neural tube. While Shroom3 mediated apical constriction occurs as part of the rolling up, cells adjacent to constricting cells are shown here to undergo apical expansion, and to be elongated in the anterior posterior axis. This behavior accommodates the elongation of the neural plate while allowing it to converge medolaterally. Modeling shows that in principle this effect can be caused in a rectangular field by apical constriction of some cells, with the overall forces of the rectangle leading to apical expansion of the apices of cells that are not actively constricting, with stretching of cells in the long orientation.

In addition, to modeling the cell behaviors, Lmo7 is used as a mediator of apical constriction precociously in the blastula stage, and is shown to induce similar flattening of cells, thus illustrating that this model could apply to the normal AP elongation of non-apically-constricting cells. The model and experiment support the authors contention that this mixture of AP elongating cells with apically constricting cells may be sufficient to allow elongation of the neural plate during neural plate closure.

However the argument does not include the likely contribution of other forces in neural tube closure, nor does it provide a satisfactory explanation for the requirement for Planar Cell Polarity signaling. For example, the flattening of cells has been discussed before, e.g. example by Hardin and Keller, who ascribed this to the stiffness of adjacent tissues.

Reviewer 3 is asking questions related to 1) the contribution of various cell behaviors to neural tube closure, 2) the origin of forces, and 3) the role of PCP signaling in the

regulation of these forces and behaviors. Our responses have been divided into these three categories.

The model does not seem to apply in the anterior neural plate, where there is less PCP dependent neural tube narrowing, and less extension. So far as I see, particularly from the movies in Butler and Wallingford also much less stretching of non constricting cells, yet the neural tube rolls up effectively. This behavior provides a good comparison and should be more effectively considered and contrasted as a different set of behaviors. I note that recent work from Baldwin and Wallingford has used careful analysis of apical expansion and constriction, and that work also needs to be compared and contrasted more thoroughly in the manuscript. Since the authors essentially dismiss a significant role of the mediolateral intercalation of the cells in the normal neural plate closure, I think it is important to review the data from Butler and their own data to see how much quantitatively it contributes to narrowing of the posterior neural plate. I don't see that this lack of importance has been presented elsewhere in the literature.

We apologize for the misleading introductory sentences that appeared to dismiss a role of convergent extension in NP narrowing. As noted by the reviewer, several recent studies analyzed different aspects of neural tube closure (Baldwin et al., 2022; Butler and Wallingford, 2018; Christodoulou and Skourides, 2022a). At the initial stage of neurulation, mediolateral cell intercalation narrows the NP. During the next stage, apical constriction becomes apparent in both the anterior and the posterior NP to initiate the formation of neural furrow/groove and then the neural fold, whereas cell intercalations become less frequent. Finally, the process completes with the neural folds fusing dorsally into the neural tube.

Our study focuses on the contribution of apical constriction to epithelial bending rather than other processes, such as the early narrowing of the NP or the fusion in the later stages of NTC. The NP narrowing has not been addressed in the manuscript, because it is not expected to promote epithelial bending on its own. As requested, we have compared our work with the other studies. Consistent with the important role of mediolateral intercalations in NP narrowing, we observe that about 25% cells are involved in intercalation events in the posterior NP during stage 13-15, which agrees well with the data from Butler and Wallingford (2018).

As we explained in response to reviewer 2, our data generally agree with the above-cited studies with respect to apically constricting cells. The main difference between our work and that of the other studies is that we separately analyzed cell behaviors in the hinge and non-hinge areas. The hinge regions are challenging to track and analyze quantitatively in time-lapse imaging, yet they show high degree of cell heterogeneity. In our analysis of both fixed and live embryos, the dorsolateral hinges revealed comparably high apical domain heterogeneity in the anterior and the posterior NP (data not shown). Since the number of the cells at the hinges is relatively small as compared to the total cell number in the NP, it is crucial to analyze them separately.

Perhaps more importantly, the authors need consideration of the other forces that drive the narrowing of the neural plate, for example the intercalation and migration of deep cells (Davidson and Keller, 1999). These deep cells rearrange dramatically. While not readily visible, they are also influenced by PCP, and have previously been assumed to exert forces to close the neural tube. so their conclusion “We propose that neural folding relies on PCP-dependent transduction of mechanical signals between neuroepithelial cells.” may well be incorrect, since these forces may largely derive from the deep neural plate cells acting mechanically on the overlying epithelium. In addition, there is proposed to be medial force from the medial migration of the neural folds and epidermis.

It is very difficult to tease apart the contributions of the different forces. Ideally, if the authors wish to argue that the movements and shapes they see are sufficient to roll up the neural tube, then this ought to be feasible in explanted tissue, perhaps plated on a pliable sheet, to study the effects of PCP more directly on the neuroepithelial cells. Despite the drawbacks, the authors make a useful contribution, but as it stands it is incomplete, both in terms of placing it in the context of the current literature and in the level of understanding of how the cells become elongated or compressed.

Related to this issue brought up by the referee, we observed that dorsal explants that were prepared at the end of gastrulation and lacked ventral and most of the endodermal tissues, developed apical domain heterogeneity and formed neural grooves. We decided not to include this experiment in the paper, but cited similar results reported by others (Christodoulou and Skourides, 2022b; Poznanski et al., 1997; Zhou et al., 2015). In amended Discussion, we state that “whereas our results are consistent with apical constriction being a driver of initial neural plate bending, we do not exclude potential contributions of the inner layer cells and forces coming from non-neural tissues to neural tube closure”.

The role of PCP is not shown to be a direct one on the epithelium, and so it is important to put that into context also.

The mechanism of how PCP signaling regulates these forces is beyond the scope of the current work. We note that mosaic depletion of Vangl2 strongly reduced apical constriction and apical domain heterogeneity in the manipulated ectoderm cells but minimally affected embryo length (Fig. S3). This suggests that Vangl2 is cell-autonomously required for intrinsic force generation. On the other hand, PCP protein signaling can also regulate the thickness of the deep cell layer underlying the superficial layer of the neural plate (Ossipova et al., 2015), an activity that may non-cell autonomously contribute to the Vangl2 depletion phenotype. PCP proteins have been reported to act non-cell autonomously during vertebrate gastrulation and neurulation (Sokol, 1996). We therefore acknowledge that both intrinsic and extrinsic PCP-dependent forces are likely to contribute to NT closure and revised the text accordingly. Also, please also see our response to reviewer 1.

Of course, it is understandable that the authors want to emphasize their own contributions, but the failure to put their work in proper context does appear as

ungenerous or selective. For example: “We conclude that the neural plate acquires considerable cell heterogeneity by the onset of folding, a result supported by another study 7. “ is simply the wrong presentation. Chronologically it is hard to see how an older study supports a newer study, and the authors need to turn the sentence around.

We apologize for this misrepresented point, the sentence in question has been properly revised.

References

Baldwin, A., Popov, I. K., Keller, R., Wallingford, J. and Chang, C. (2023). The RhoGEF protein Plekhhg5 regulates medioapical and junctional actomyosin dynamics of apical constriction during *Xenopus* gastrulation. *Mol Biol Cell* **34**, ar64.

Baldwin, A. T., Kim, J. H., Seo, H. and Wallingford, J. B. (2022). Global analysis of cell behavior and protein dynamics reveals region-specific roles for Shroom3 and N-cadherin during neural tube closure. *Elife* **11**.

Butler, M. T. and Wallingford, J. B. (2018). Spatial and temporal analysis of PCP protein dynamics during neural tube closure. *Elife* **7**.

Christodoulou, N. and Skourides, P. A. (2022a). Distinct spatiotemporal contribution of morphogenetic events and mechanical tissue coupling during *Xenopus* neural tube closure. *Development* **149**.

Christodoulou, N. and Skourides, P. A. (2022b). Somitic mesoderm morphogenesis is necessary for neural tube closure during *Xenopus* development. *Front Cell Dev Biol* **10**, 1091629.

Davidson, L. A. and Keller, R. E. (1999). Neural tube closure in *Xenopus laevis* involves medial migration, directed protrusive activity, cell intercalation and convergent extension. *Development* **126**, 4547-56.

Hirano, S., Mii, Y., Charras, G. and Michiue, T. (2022). Alignment of the cell long axis by unidirectional tension acts cooperatively with Wnt signalling to establish planar cell polarity. *Development* **149**.

Holcomb, M. C., Gao, G. J., Servati, M., Schneider, D., McNeely, P. K., Thomas, J. H. and Blawdziewicz, J. (2021). Mechanical feedback and robustness of apical constrictions in *Drosophila* embryo ventral furrow formation. *PLoS Comput Biol* **17**, e1009173.

Mancini, P., Ossipova, O. and Sokol, S. Y. (2021). The dorsal blastopore lip is a source of signals inducing planar cell polarity in the *Xenopus* neural plate. *Biol Open* **10**.

Matsuda, M., Chu, C. W. and Sokol, S. Y. (2022). Lmo7 recruits myosin II heavy chain to regulate actomyosin contractility and apical domain size in *Xenopus* ectoderm. *Development* **149**.

Ossipova, O., Chu, C. W., Fillatre, J., Brott, B. K., Itoh, K. and Sokol, S. Y. (2015). The involvement of PCP proteins in radial cell intercalations during *Xenopus* embryonic development. *Dev Biol* **408**, 316-27.

Poznanski, A., Minsuk, S., Stathopoulos, D. and Keller, R. (1997). Epithelial cell wedging and neural trough formation are induced planarly in *Xenopus*, without persistent vertical interactions with mesoderm. *Dev Biol* **189**, 256-69.

Sokol, S. Y. (1996). Analysis of Dishevelled signalling pathways during *Xenopus* development. *Current Biology* **6**, 1456-1467.

Sweeton, D., Parks, S., Costa, M. and Wieschaus, E. (1991). Gastrulation in *Drosophila*: the formation of the ventral furrow and posterior midgut invaginations. *Development* **112**, 775-89.

Xie, S. and Martin, A. C. (2015). Intracellular signalling and intercellular coupling coordinate heterogeneous contractile events to facilitate tissue folding. *Nat Commun* **6**, 7161.

Zhou, J., Pal, S., Maiti, S. and Davidson, L. A. (2015). Force production and mechanical accommodation during convergent extension. *Development* **142**, 692-701.

REVIEWERS' COMMENTS

Reviewer #1 (Remarks to the Author):

The points which I raised have been adequately addressed and I am satisfied with the responses of the authors.

Inability of direct estimation of force by laser ablation is unfortunate but I hope that the authors will be able to achieve it and prove the pulling force someday.

Reviewer #2 (Remarks to the Author):

The authors addressed all of my concerns.

Reviewer #3 (Remarks to the Author):

This revised manuscript more clearly presents the arguments for new findings on morphogenesis of the neural plate, specifically the way that stochastic apical constriction, separated by cells which expand their apices in the AP axis causes anisotropic folding of the neural plate. The observations are clear, and tested experimentally also with other mediators of apical constriction that are ectopically expressed in the animal cap ectoderm. The model provides a satisfying alternative to the idea that cell junctions which contract anisotropically might drive folding, though superficially the result is the same, namely that a lot of the contracting cells still have anisotropic apices lengthened in the AP axis.

The other intriguing part of the manuscript is the requirement for PCP proteins in this process. VAngl2 and Fuzzy (not shown) have the same effect of eliminating the anisotropic appearance of cells, and also reduces the amount of junctional actin. This is an intriguing result which does not fit readily into models of how PCP proteins work in polarizing cells, but is not developed further.

Overall, the revised manuscript makes a useful contribution to the increasing understanding of neural tube closure,

Although reviewer 1 raised the general question of external forces in elongation, the authors only tests vangl2 knockdown in the ectoderm. It would have been useful to isolate the changes by knockdown in the mesoderm only, and examine the effect on the neural plate when the underlying mesoderm fails to elongate properly, but I am not going to make a big deal out of this.

The additional changes to the presentation have clarified the arguments.

A point-by-point response to the reviewers' comments

We thank all three reviewers for their thorough and constructive comments that helped us improve the manuscript. The following statements have been included in the revised manuscript.

Reviewer #1 (Remarks to the Author):

Inability of direct estimation of force by laser ablation is unfortunate but I hope that the authors will be able to achieve it and prove the pulling force someday.

We included the following statement in Discussion. "Currently, lack of more direct measurement of tension by laser ablation is limiting the significance of our findings."

Reviewer #3 (Remarks to the Author):

Although reviewer 1 raised the general question of external forces in elongation, the authors only tests vangl2 knockdown in the ectoderm. It would have been useful to isolate the changes by knockdown in the mesoderm only, and examine the effect on the neural plate when the underlying mesoderm fails to elongate properly, but I am not going to make a big deal out of this.

We included the following statement in Discussion. "Whereas our work focuses on the forces originating in the superficial neural plate, it would be of interest to determine the effect of mesoderm-specific Vangl2 knockdown on the apical domain heterogeneity and bending of the NP."